# An optogenetic-phosphoproteomic study reveals dynamic Akt1 signaling profiles in endothelial cells

Wenping Zhou [1,2,3,7], Wenxue Li [1,4,7], Shisheng Wang [5,7], Barbora Salovska [1,4], Zhenyi Hu [1,4], Bo Tao [1,3], Yi Di [1,4], Ujwal Punyamurtula [6], Benjamin E. Turk [1], William C. Sessa [1,3] ✉ & Yansheng Liu [1,4] ✉

The serine/threonine kinase AKT is a central node in cell signaling. While aberrant AKT activation underlies the development of a variety of human diseases, how different patterns of AKT-dependent phosphorylation dictate downstream signaling and phenotypic outcomes remains largely enigmatic. Herein, we perform a systems-level analysis that integrates methodological advances in optogenetics, mass spectrometry-based phosphoproteomics, and bioinformatics to elucidate how different intensity, duration, and pattern of Akt1 stimulation lead to distinct temporal phosphorylation profiles in vascular endothelial cells. Through the analysis of ~35,000 phosphorylation sites across multiple conditions precisely controlled by light stimulation, we identify a series of signaling circuits activated downstream of Akt1 and interrogate how Akt1 signaling integrates with growth factor signaling in endothelial cells. Furthermore, our results categorize kinase substrates that are preferably activated by oscillating, transient, and sustained Akt1 signals. We validate a list of phosphorylation sites that covaried with Akt1 phosphorylation across experimental conditions as potential Akt1 substrates. Our resulting dataset provides a rich resource for future studies on AKT signaling and dynamics.

Dynamic patterns of protein phosphorylation dictate the output of cell signaling networks. A key example is the protein kinase AKT, a central node in signaling networks regulating multiple cellular processes, including cell growth, migration, survival, metabolism, and angiogenesis[1,2]. Growth factor and insulin stimulation lead to activation of class I phosphoinositide 3-kinase (PI3K), which generates the lipid second messenger phosphatidylinositol-3,4,5-trisphosphate [PI(3,4,5)P$_3$, or PIP3]. AKT is recruited via its PH domain to PIP3 and phosphatidylinositol-3,4-bisphosphate [PI(3,4)P2] in the plasma membrane (PM)[3–6]. The recruitment of AKT to PM promotes its

phosphorylation by phosphoinositide-dependent kinase-1 (PDK1) and mechanistic target of rapamycin complex-2 (mTORC2) at residues Thr308 and Ser473[7,8], which are essential for AKT catalytic activity. In addition, recent evidence suggests that binding to phosphoinositides relieves autoinhibition of the catalytic domain by the PH domain, suggesting that AKT is only active and capable of phosphorylating substrates when membrane-bound[5,6,9]. Alternatively, it has been proposed that mTORC2 phosphorylation alone can relieve these autoinhibitory constraints[10,11]. Interestingly, the temporal patterns of AKT activation have been suggested to selectively regulate metabolic

[1]Department of Pharmacology, Yale University School of Medicine, New Haven, CT 06510, USA. [2]Department of Cell Biology, Yale University School of Medicine, New Haven, CT 06511, USA. [3]Vascular Biology & Therapeutics Program, Yale University School of Medicine, New Haven, CT 06520, USA. [4]Cancer Biology Institute, Yale University School of Medicine, West Haven, CT 06516, USA. [5]Department of Pulmonary and Critical Care Medicine, and Proteomics-Metabolomics Analysis Platform, West China Hospital, Sichuan University, Chengdu 610041, China. [6]Master of Biotechnology ScM Program, Brown University, Providence, RI 02912, USA. [7]These authors contributed equally: Wenping Zhou, Wenxue Li, Shisheng Wang. ✉e-mail: william.sessa@yale.edu; yansheng.liu@yale.edu

responses to insulin stimulation[12] and to selectively provoke specific downstream phosphorylation events[13]. In a transgenic mouse model, short-term (2 weeks) Akt1 activation in cardiac muscle cells leads to enhanced angiogenesis, whereas longer-term (6 weeks) Akt1 activation by contrast reduces angiogenesis[14]. In addition, the loss of Akt1 globally[15] or selectively in endothelial cells[16] reduces angiogenesis in some models while enhancing it in others[17]. How cells decode the same signaling input from Akt1 into distinctive downstream signaling remains largely unknown[18].

Studying dynamic regulation of AKT with "natural" stimuli like growth factors is difficult as one cannot control AKT activity with temporal precision. Also, those stimuli necessarily activate many other pathways, making it impossible to isolate the contribution of AKT alone. Optogenetics, based on genetically encoded light-controllable proteins, allows for the precise control of signaling processes[19,20], and thus offers an alternative, powerful tool for studying temporal patterns of AKT activation[21–23]. For example, Xu et al. described an optogenetic tool, Opto-Akt1[23], based on the *Arabidopsis* CIB1 and cryptochrome 2 (CRY2) proteins. Blue-light illumination induces heterodimerization of a CRY2-Akt1 fusion with the N-terminus of CIB1 (CIBN) that was anchored to the membrane by prenylation, thereby recruiting the Akt1 to the PM to promote Akt1 phosphorylation. Kawamura et al. recently used a similar opto-Akt2 construct to reveal Akt2-induced changes in metabolic pathways in skeletal muscle cells[24]. Use of these optogenetic systems has provided insights into how patterns of AKT signaling impact cellular protection against oxidative stress[22] and neuronal damage[25]. However, investigation of signaling output in these studies was mostly limited to phosphorylation of AKT itself and a few of its well-established targets by immunoblotting.

One way to provide a more global view of signaling output from a specific pathway would be to combine optogenetic tools with mass spectrometry (MS)-based phosphoproteomics[20]. So far, reported studies doing so utilized traditional shotgun MS methods with a limited number of biological conditions[26,27]. As an emerging MS-based technique, data-independent acquisition (DIA)-MS[28,29] offers sensitive and reproducible measurement of endogenous levels of proteins and integrated proteomes[30–33]. Recent software developments for DIA-MS have further enabled the analysis of post-translational modifications (PTM), such as phosphorylation[34–36]. Because DIA quantification is accomplished at the peptide-fragment level, unique MS/MS ion signatures can be used to support the precise PTM site-level detection, localization, and quantification across a large number of samples[34,35,37]. In the present study, we have integrated a phosphoproteomics-DIA workflow[38,39] with an established optogenetic system to evaluate how Akt1 dynamically impacts the cellular phosphoproteome through dozens of different temporal patterns of stimulation in endothelial cells (EC). Our results reveal distinct phosphorylation circuits downstream of Akt1 signaling and provide a unique resource for future studies focusing on AKT.

## Results

### Developing Optop-DIA for studying Akt1 signaling

To investigate the phosphoproteomic consequences of AKT signaling and temporal dynamics, we developed an Optogenetics-phosphoproteomics DIA-MS experimental system (hereafter, Optop-DIA) with the following configurations and considerations.

Firstly, to achieve precise temporal control of Akt1 phosphorylation levels, we adopted a lentiviral, optogenetic construct[23] that co-expresses mCherry-CRY2-Akt1 with a CAAX motif (plasma membrane anchor) fused to the C terminus of CIBN-GFP (Fig. 1a & Supplementary Fig. 1). Strong Akt1 phosphorylation following blue light exposure was confirmed by immunoblotting for phospho-T308 (or pT308) and pS473 of Akt1 (Fig. 1b). Spinning disk confocal microscopy indicated that translocation of mCherry-CRY2-Akt1 to the plasma membrane after blue light stimulation was rapid (within seconds) (Fig. 1c). Stable

p-Akt1 levels upon continuous light exposure were observed at 5 min and lasted up to 30 min or longer (Supplementary Fig. 2a–c). Shutting off the light led to efficient dephosphorylation of Akt1 and its substrate eNOS in about 10 min and almost eliminated their phosphorylation at 30 min (Fig. 1c, d and Supplementary Fig. 2b). Furthermore, inhibition of PI3K with LY294002 (10 μM) diminished light-induced Akt1 phosphorylation levels (Supplementary Fig. 2c), demonstrating that, similar to endogenous Akt1, the phosphorylation of the opto-AKT requires endogenous PI3K activity. These results show that the optogenetic Akt1 system can be efficiently controlled by manipulating the light source.

Subsequently, we selected suitable cells and light conditions for perturbing Akt1 phosphorylation. Previously, the three isoforms of AKT (Akt1, Akt2, and Akt3) were reported to have variable expression patterns and variable impacts on the phosphoproteome[40]. Akt1 is the predominant isoform expressed in endothelial cells (EC)[16], and we have shown that Akt1 exerts a non-redundant function during physiological angiogenesis in EC. We thus focused on Akt1 signaling in EC, in which Akt2 likely plays a minor role[16]. We selected EA.hy 926, a well-established human vascular EC line, for optogenetic experiments. We found by immunoblotting that increasing the light intensity from 0.01 to 1 mW/cm² led to a stepwise increase in Akt1 phosphorylation, as well the downstream phosphorylation of the Akt1 substrates eNOS and GSK3β (Fig. 1e). Intensities greater than 1 mW/cm² did not further increase the level of Akt1 phosphorylation, and importantly, light activation did not induce endogenous Akt1 or ERK phosphorylation (Supplementary Fig. 2d). Thus, for MS analysis, we chose an intensity of 0.25 mW/cm² as the highest light exposure level. As previously described, this level of light did not cause long-term phototoxicity[22,41]. Parental (non-transduced) EA.hy 926 cells and HUVECs, when stimulated with 0.25 mW/cm² blue light for 30 min, showed no detectable phosphoproteomic variance or loss of cell viability (Supplementary Fig. 3a, b). Finally, to test the functional relevance of the system, we measured nitric oxide (NO) release, since AKT phosphorylation of eNOS on Ser1177 in EC promotes NO release. Indeed, 1-h light illumination at 0.25 mW/cm² significantly increased cellular NO production compared to cells kept in the dark (Supplementary Fig. 2e). Taken together, an optogenetic Akt1 system was optimized, which may help interrogate Akt1 function in endothelial cells.

Based on the above results, we applied light of 0.05, 0.10, and 0.25 mW/cm² (or 005/010/025 in short), representing minimal, modest, and high Akt1 activation levels, separately to transduced EA.hy 926 cells (Fig. 1f). By controlling light, we included three patterns of Akt1 activation: Sustained (Su), Periodic (Pe, in which cells are exposed to light for 5 min and then kept in the dark for another 5 min, in a repeated fashion), and Pulsed (Pu, 5 min light illumination followed by darkness). We focused our measurements in the first 30-min window, sampling at 10 min intervals. We observed no global changes to the proteome in the transduced optogenetic cells after blue light treatment under conditions of highest intensity and duration (i.e., 0.25 mW/cm² for 30 min) (Supplementary Fig. 3c, d). This was expected, because changes in protein expression detectable by routine global proteomic analysis normally take several hours[33,42]. Accordingly, we did not measure the total proteome in subsequent experiments, because the impact of changes in protein levels on phosphoproteomic readouts would be minimal within a 30 min timeframe. Collectively, we used 27 conditions of Akt1-activation (3 light intensities × 3 patterns × 3 time points), together with full darkness conditions as a control, for phosphoproteomic analysis, with all conditions examined in biological duplicates.

Using phosphoproteomics-DIA[38,39], together with a spectral-library-free directDIA algorithm[43,44], we identified a total of 37,449 unique Class-I phosphopeptides (with confident phosphorylation site-localization)[35,45], corresponding to 5044 phosphoproteins, in the entire experiment (both peptide and protein FDRs were kept below 1% by

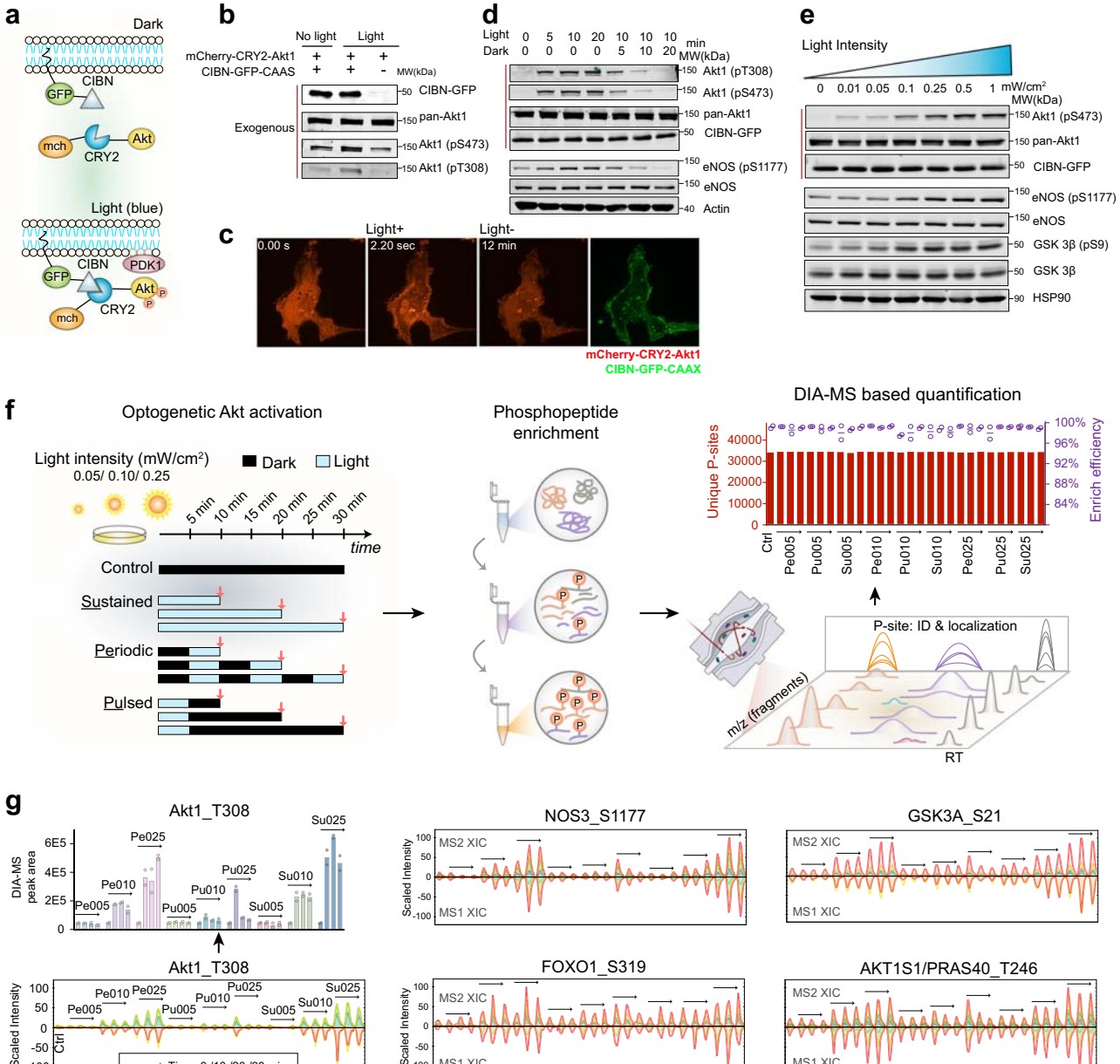

**Fig. 1 | The optogenetic Akt1 system and experimental design of Optop-DIA.**
**a** The optogenetic Akt1 system upon blue light activation. **b** Increased Akt1 phosphorylation after light illumination with co-expressed mCherry-CRY2-Akt1 and CIBN-GFP-CAAX constructs in HeLa CCL2 cells (Light intensity, 16 mW/cm²). **c** Live-cell imaging demonstrating that upon about 2 seconds blue light (488 nm laser) stimuli, Akt1 translocates to the plasma membrane. Green channel validating the CIBN-GFP-CAAX expression and location. **d** Light-induced Akt1 phosphorylation in optogenetic EA.hy 926 cells from 5 to 10 min and upon removal of light (light intensity, 2 mW/cm²). **e** Western blot of light intensity 0.01, 0.05, 0.1, 0.25, 0.5 to 1 mW/cm² induced Akt1 phosphorylation in transduced EA.hy 926 cells. In (**b**), (**d**), and (**e**), the exogenous Akt1 and p-Akt1 signals are marked by dark red vertical line. **f** Systemic graph showing phosphoproteomic experimental design with light blue bars indicating light exposure and black bars for dark. Phosphopeptide enrichment and phosphoproteomics-DIA were performed on collected cell lysates, resulting in excellent consistency of sensitivity between different samples among Akt1

activation groups. The peptide fragment level information is used in DIA-MS for phosphosite (P-site) detection and quantification. Center bars denote mean of phosphopeptide enrichment efficiency percentage in the right panel. **g** Akt1 pT308 across all the samples and Akt1 substrates phosphorylation levels profiled by Optop-DIA. Left panel: The extracted ion chromatography (XIC graph) from DIA analysis for pT308 is used to infer the quantitative changes between samples. In the XIC, the peak groups above the middle line denote the MS2 level ion traces (i.e., peptide fragments) for the phosphopeptide signature, whereas the peak groups below the middle line denote the MS1 level ion traces (i.e., isotopic m/z of the intact phosphopeptide). Middle and right panels: The XIC graphs for the other four Akt1 substrates based on respective phosphopeptide signatures. The arrows indicate the identical Akt1 activation conditions as illustrated for Akt1 pT308 in the left panel. For (**f**, **g**), $n = 2$ biologically independent samples. For (**b**, **d**, **e**), two times of WB experiments were repeated independently with similar results. Source data are provided as a Source data file.

Spectronaut[44]). This translates to 34,740 phosphorylation sites (P-sites) that were confidently identified, unambiguously localized, and quantified in all Akt1-activation conditions (Fig. 1f, right panel, also see "Methods"). Similar to our previous phosphoproteomics-DIA reports[38], biological duplicates yielded Pearson correlations as high as $R = 0.943$

on average (Supplementary Fig. 4a). Phosphopeptide enrichment efficiency reached $98.4 \pm 0.5\%$ across all samples.

Next, we inspected DIA quantitative profiles for p-Akt1. We quantified the relative level of Akt1 pT308 across all the samples and conditions using extracted ion chromatography (XIC graph) at the

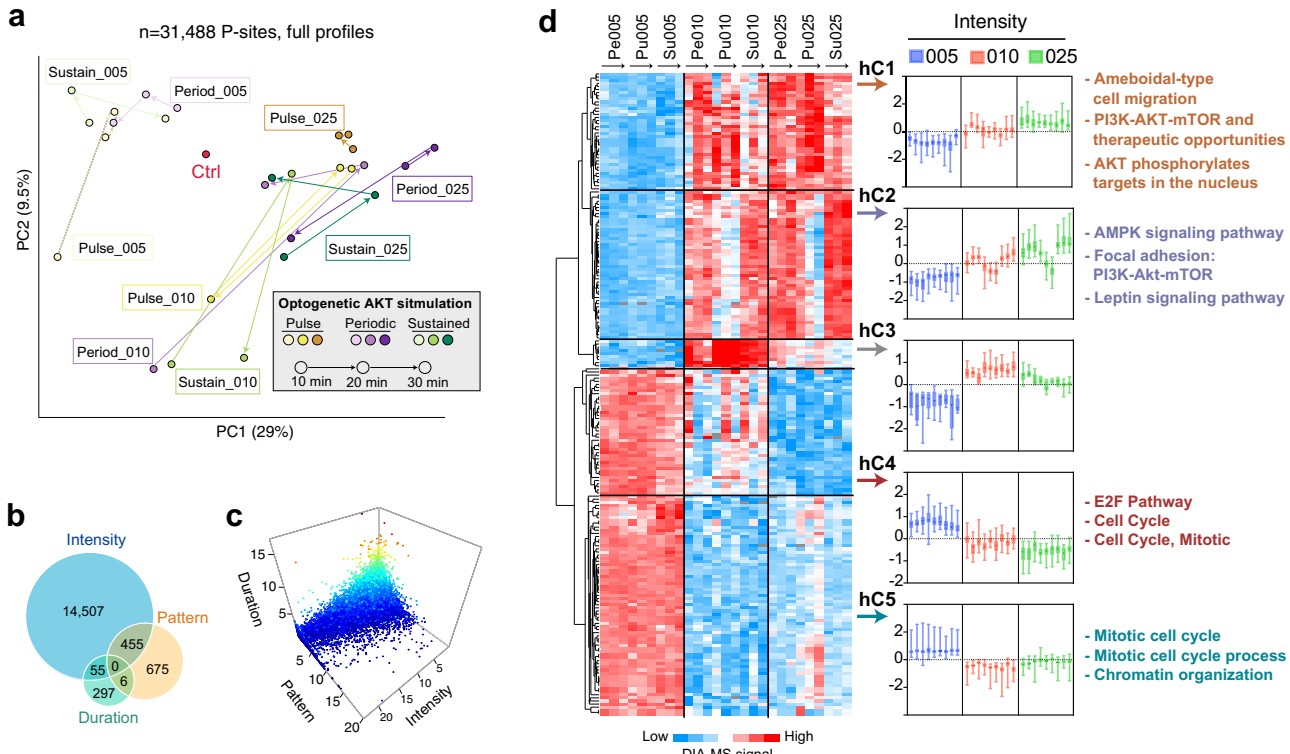

**Fig. 2 | Global variability of Akt1 responsive P-sites across all conditions driven by Akt1 intensity. a** Principal component analysis (PCA) based on all conditions using 31,488 P-sites quantified with full profiles without missing values. **b** The number of differential P-sites filtered by linear ANOVA analysis that considers co-variables for Intensity, Pattern, and Duration traits. All 34,740 P-sites were analyzed in this analysis. Venn diagram visualizes the size and overlapping identities following ANOVA. **c** The 3-dimensional (3D) plot of P-values projected to the traits of Intensity, Pattern, and Duration based on ANOVA. Each dot here represents a particular P-site. The x-, y-, and z-axes of the P-site denote the P values in the Log2 scale. The blue-to-red color visualizes the low-to-high Log2(P value) when the Duration trait is considered as the major varying factor. P values were based on linear ANOVA analysis (two-sided, unadjusted). **d** Heatmap for results of paired Student's T-test for intensity groups, deconvoluting five groups (clusters hC1-hC5) based on phosphorylation and dephosphorylation trend. The blue-to-red color bar denotes the centered DIA-MS quantitative signals. The GO processes and keywords on the right indicate representative items enriched in hC1-hC5 by Metascape annotation. No GO items were enriched in hC3 due to its small size. The log2 fold-changes of each time points to control are visualized by boxplots (center line, median; box limits, upper and lower quantiles; whiskers, Min to Max values). $n = 38$, 48, 9, 41, and 71 observations of P-sites in hC1-hC5 boxplots). Source data are provided as a Source data file.

MS2 level (Fig. 1g, left). XICs showed 3.4- and 7.1-fold upregulation of Akt1 activity in the 010 and 025 groups, respectively, as compared to the 005 group. Reassuringly, profiles in pulsed experiments confirmed the timely removal of Akt1 pT308. We also succeeded in quantifying Akt1 pS473 in ~90% of the samples and observed a similar profile (Supplementary Fig. 4b). Well-characterized Akt1 downstream P-sites[1], such as NOS3 pS1177, GSK3A pS21, FOXO1 pS319, and AKT1S1 pT246, all showed expected regulatory patterns across conditions (Fig. 1g). More examples are provided in Supplementary Fig. 4b. Thus, desirable Akt1 regulation was achieved by optogenetics and then captured by DIA-MS.

In summary, our Optop-DIA workflow harnesses an efficient optogenetic system with a high-performance phosphoproteomics-DIA technique for investigating Akt1 signaling.

**AKT-responsive P-sites are pervasive and strongly regulated by the intensity of stimulation**

To assess the global quantitative landscape of Akt1-mediated protein phosphorylation, we performed principal component analysis (PCA) based on 31,488 P-sites quantified with full profiles across all conditions (Fig. 2a). With two principal components, conditions under minimal light intensity (005) can be well-separated from all conditions under moderate and high light exposure (010 and 025), irrespective of light pattern and duration. Furthermore, within each intensity group, neither the light *duration* (10/20/30 min) nor *pattern* (Pe/Pu/Su) could

globally classify samples into obvious clusters. Hierarchical clustering analysis (HCA) supported this observation (Supplementary Fig. 4c). Next, to substantiate PCA and HCA, we deployed a special ANOVA analysis[46] for all 34,740 P-sites, in which one of the *intensity*, *pattern*, or *duration* variables was kept as the major factor with the other two as co-variables (see "Methods"). This analysis verified that *intensity* has the most profound impact among the three variables. Using the same criteria ($P < 0.01$), 15,017, 1136, and 358 P-sites were identified to be significantly associated with the *intensity*, *pattern*, and *duration* variables respectively, corresponding to 43.2%, 3.27%, 1.03% of the total quantified phosphoproteome (Fig. 2b). The setting of co-variables in this ANOVA test[46] renders very few P-sites (i.e., <1.5%) assigned as responsible to two traits, and drastically lower P-values were obtained if *intensity* was considered as the major variable factor (Fig. 2c). Taken together, Akt1 perturbations could influence pervasive P-sites, most of which can be ascribed to variability in the level of p-Akt1.

To decipher the *intensity*-driven biological processes, we extracted the top P-sites differential between the 005/010/025 *intensity* groups using paired student's *t*-tests, since the nine conditions across each group are exactly matched if the *intensity* is perceived as the variable factor. A total of 207 P-sites were filtered ($P < 0.01$ between three groups, fold change > 1.5 in the 010 vs. 005 and 025 vs. 005 comparisons), deconvoluting P-site profiles into five characteristic clusters (Fig. 2d, see Supplementary Fig. 5 for protein names and their potential interactions). While the first three clusters (hC1-hC3)

generally show P-site up-regulation upon higher light exposure, a 3rd minor cluster (hC3, $n = 9$) harbors salient dephosphorylation in the O25 group compared to the O10 group. Interestingly, the profile divergence in the pulsed stimulation essentially distinguishes hC1 and hC2: in both O10 and O25 groups, after 5-min light illumination, P-sites in hC2 ($n = 48$) immediately diminished upon p-Akt1 removal, whereas robust up-regulation persisted for at least 30 min for P-sites in hC1 ($n = 38$). As expected, p-T308 of Akt1 was classified into hC2. Protein-level annotation based on Metascape[47] uncovered that while hC1 and hC2 did share many biological processes, hC1 uniquely enriched for AKT targets in the nucleus, while hC2 enriched pathways such as AMPK signaling and leptin signaling (Fig. 2d, all $P$ values < $10^{-8}$, *hypergeometric* test by Metascape). Given that the active pool of Akt1 in the optogenetic system is likely restricted to the PM, the observed enrichment for Akt1 targets in the nucleus suggests that they in fact become phosphorylated in the cytoplasm as has been proposed by others[48]. The clusters hC4 ($n = 41$) and hC5 ($n = 71$), that tended to become dephosphorylated upon Akt1 activation were mainly enriched for cell cycle and mitotic processes ($P < 10^{-9}$), pointing to indirect regulation of cell proliferation.

In summary, although the extent of Akt1 activation (i.e., *intensity*) appears to determine global phosphoproteomic variability, paired statistical analysis suggests that increasing light does not always lead to proportional increases in downstream P-sites.

## Regulated phosphorylation events with variable decay following short-term Akt1 activation

Because pulsed stimulation appears to elicit Akt1 decay-dependent phosphorylation, we further analyzed different temporal signaling profiles per P-site. After short-term optogenetic activation, 5-min of darkness reduced p-Akt1 levels by only ~50–60% rather than eliminating it (see Fig. 1d). Thus, periodic illumination effectively reduced and then restimulated p-Akt1 levels at 10, 20, and 30 min. To focus on those P-sites most responsive to Akt1, we filtered 1873 Akt1-activated P-sites that were repeatedly up-regulated by >1.5 fold in each of the three periodic activations in the Pe025 group. Subsequently, we performed fuzzy c-means clustering (FCM) analysis on these P-sites, but now in the pulsed condition (i.e., Pu025). FCM classified these P-sites into four clusters (fC1-fC4), each exhibiting different behavior (Fig. 3a, b, and Supplementary Fig. 6a for an averaged profile per cluster). The first cluster, fC1, contained Akt1 pT308 and P-sites that changed rapidly in response to modulation of p-Akt1 levels. The second cluster, fC2, represeneds P-sites that decreased following p-Akt1 removal, with delayed kinetics in comparison to fC1. The third cluster, fC3, showed sustained phosphorylation albeit with limited decreases over the 30 min timeframe. The fourth cluster, fC4, demonstrated graded upregulation over the full-time course (Fig. 3b). Together, these results underscore time-dependent, sequential phosphorylation events in Akt1 signaling.

To compare the sequence features and functions of P-sites among clusters fC1-fC4, we initially performed motif enrichment analysis using motifeR[49] (Fig. 3c). The Akt1 substrate motif has been described as R-x-R-x-x-S*/T*[50,51], with x corresponding to any amino acid, and S*/T* representing the phosphorylated serine or threonine residue. In keeping with potential enrichment for direct Akt1 substrates, we found this motif overrepresented in fC1, along with a more generic R-x-x-S*/T* motif that would include Akt1 substrates among those of many other kinases. Furthermore, the classic MAPK (ERK, p38, JNK) motif P-x-S*/T*-P was overrepresented in fC3, and fC4 enriched a longer version of this motif, P-x-S*/T*-P-x-P[52] as well as the favored motif for CK2, S*/T*-D-x-E[53]. Although cluster fC1 has the least P-sites among clusters ($n = 189$), 16% of them carry the strict Akt1 motif (Fig. 3d), about half of which are established Akt1 P-sites (Supplementary Data 1). In contrast, the remaining clusters fC2, fC3, and fC4 were not apparently enriched for sites matching the Akt1 motif (2.0%, 3.6%, and 1.7% of sites respectively,

compared with 3.0% for the entire dataset). We performed a P-site-level annotation and enrichment analysis against kinase/P-site relationships and regulatory processes cataloged in both the PhosphoSitePlus database (PSPdb)[54] and OmniPath resources[55]. While a series of categories, including substrates of AKT and PRKACA (PKA), were enriched in fC1 (Fig. 3e), fC2 and fC3 showed moderate enrichment only for EGF-induced and RPS6KA1 substrates ($P = 0.0369$ and 0.0178, Fisher's exact test). Notably, fC4 was enriched in P-sites for CK2α1 (CSNK2A1, $P = 0.0478$), CDK1 ($P = 0.0183$), and CDK2 ($P = 0.0316$). These results largely align with motif analysis. Finally, we performed an integrative phosphoproteomic data analysis to further understand the sequential signaling events following short-term p-Akt1 activation by leveraging recently developed phosphoproteomics-focused bioinformatic tools such as the signaling reconstruction tool PHONEMeS[56] and the kinase-substrate prediction tool PhosR[57]. With dynamic profiles of canonical substrates of kinases in all conditions across our experiments considered by PhosR, PHONEMeS identifies a path that connects the regulated P-sites in fC1-fC4 with co-regulated kinases[58]. The reconstructed network (Fig. 3f) suggests that short-term Akt1 stimulation activated a set of putative kinases with variable temporal patterns largely consistent with those identified in Fig. 3e. Together, transient phosphorylation of Akt1 substrates appears sufficient to induce sustained phosphorylation of unique sets of P-sites, which are potential substrates for MAPKs, CK2, and CDKs.

To summarize, in our Optop-DIA data from the pulsed condition delineates that short-term Akt1 stimulus leads to temporal processes carrying respective phosphorylation functions in a highly organized manner.

## Strength, duration, and stability of Akt1 signaling distinctly impact downstream P-sites

To further visualize the overall features of Pu/Pe/Su patterns, we expanded FCM analysis from pulsed conditions only to all illumination regimes under high light exposure. Instead of four clusters as in Fig. 3, we set FCM to categorize all Akt1-stimulated P-sites into 12 clusters (afC1-afC12, Fig. 4, and Supplementary Fig. 6b for an averaged profile per cluster), which accordingly displayed profound variability across conditions. We found that the different temporal patterns of Akt1 activation mediated the phosphorylation of the same downstream P-sites with variable dynamic profiles. The exceptional cluster is afC10, which showed almost identical up-regulation patterns among the three regimes but only represents 11.9% of all P-sites examined. The diversity of temporal profiles was also reflected in the comparable numbers of P-sites among clusters (Fig. 4). Three clusters identified with remarkable kinase-substrate enrichment were (a) afC5, enriched for AKT substrates ($P = 1.17e-11$, Fisher's exact test), (b) afC8, enriched for MYLK3 and ROCK1 substrates ($P = 3.22e-15$ and $2.26e-10$), and (c) afC9, enriched for CDK1 and CDK2 substrates ($P = 0.0048$ and $0.0256$). We found that, in concordance with fC1 in Fig. 3, afC5 showed a significant decrease of P-site abundances in the pulsed condition and was enriched in early responsive Akt1 substrates. Another cluster, afC11, exhibited a similar trend to afC5. Indeed, the percentage of P-sites with the Akt1 motif was 30.1% and 12.7% for afC5 and afC11 (Supplementary Fig. 7a) respectively, comprising 61.4% of Akt1 motif carrying peptides across all 12 clusters. On the other hand, afC9 reinforces the observation for fC4 (Fig. 3) that CDK1 and CDK2 substrate phosphorylation were sustained in the pulsed condition. However, afC9 further suggests that CDK substrates could become dephosphorylated if p-Akt1 is kept constant (Su) or oscillating at high levels (Pe). As for afC8, the 5-min periodic Akt1 stimulation tended to increase P-site abundances to a level even higher than sustained illumination. This suggests that many afC8 P-sites might favor repeated (rather than constant) Akt1 activation. P-sites with similar trends can also be found in afC1. As for

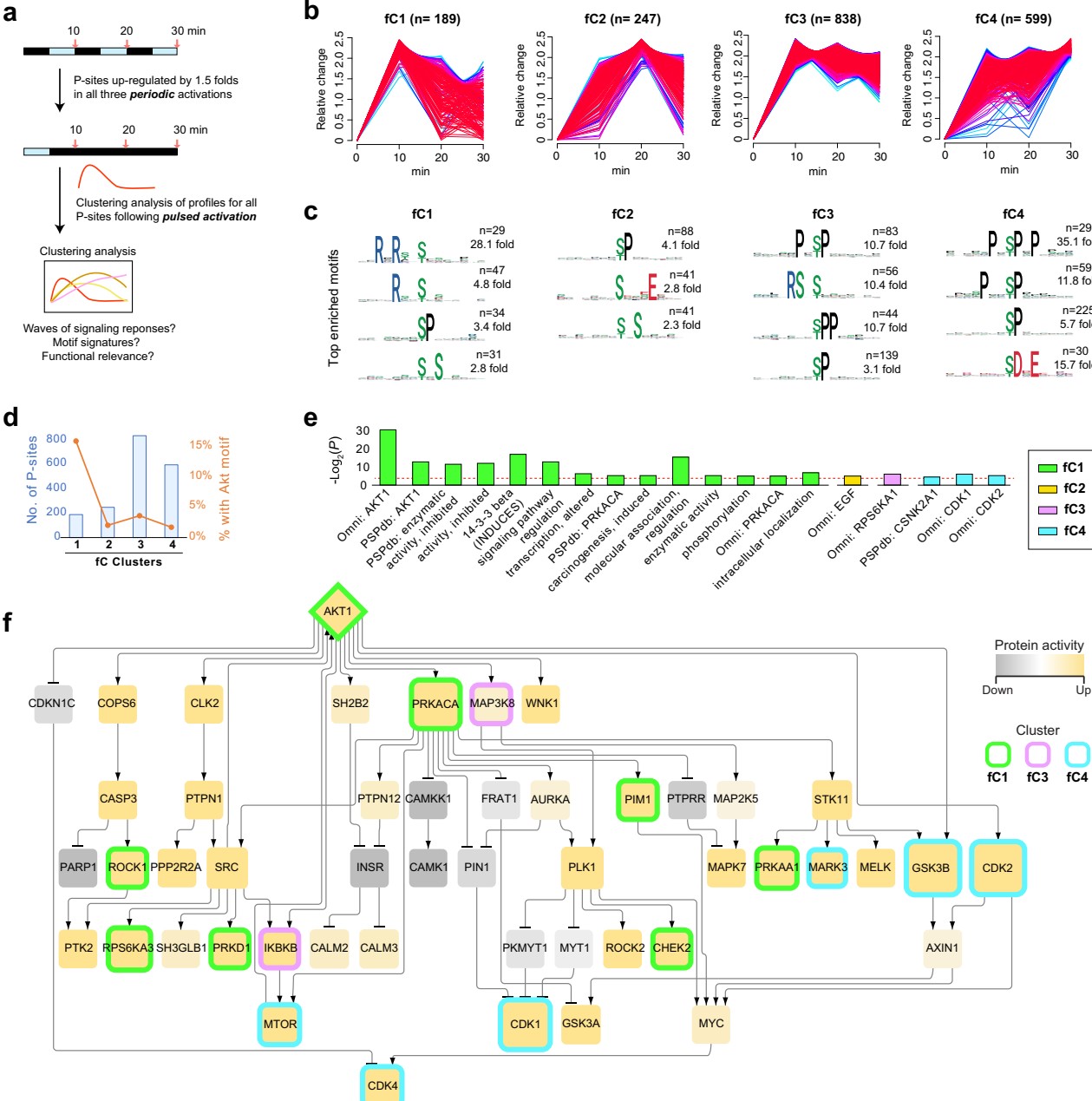

**Fig. 3 | The pulsed activation of Akt1 reveals sequential signaling events and functional clusters with variable longitudinal endurance. a** The rationale of the clustering analysis performed, which is based on the results of periodic and pulsed light activations. **b** The fuzzy c-means clustering (FCM) analysis on these P-sites and four groups (fC1-fC4) clustered. The yellow-green-purple-red color palette corresponds to the low to high membership value for each P-site, a similarity score of vectors to each cluster reported by FCM analysis. **c** The sequence features and functions of P-sites among fC1-fC4 extracted by motifeR. The number of P-sites for each motif sequence and the fold enrichment compared to the background of the entire proteome background were shown. **d** The numbers of P-sites of fC1-fC4 were shown in histograms, with the secondary axis (and the orange line) showing

different percentages of Akt1 strict motifs in fC1-fC4. **e** P-site-level annotation and enrichment analysis against kinase-P-site relationships and regulatory processes in fC1-fC4. *P* values are reported by Fisher's exact test (see "Methods"). **f** The signaling network was reconstructed using PHONEMeS, and the Akt1 subnetworks were extracted by selecting 3-step downstream nodes of Akt1. The node fill color corresponds to the protein activity inferred by PHONEMeS, with yellow and gray marking up- and downregulated activity, respectively. The border color corresponds to the cluster (fC1-fC4) in which a kinase was predominantly over-represented based on the lowest *P*-value. The node size corresponds to the number of outgoing edges. Source data are provided as a Source data file.

other clusters, afC7 showed a reciprocal profile to afC8 in which periodic stimulation induced a less profound upregulation than sustained condition. Additionally, the P-sites in afC6 were relatively suppressed after 10 min as cells acclimated to sustained light but not in the other two patterns of stimulation. Similarly, in afC12, the relative suppression by sustained activation only occurs at the earlier time points. Finally, the protein-level functional annotation

identified a few processes that are more relevant to particular clusters (Supplementary Fig. 7b, c). For example, TOR signaling appeared to be more enriched in afC11, implying that these sites did not decay immediately as seen in the afC5 cluster.

In summary, our results indicate that EC could sense variable Akt1 activation dynamics and translate them into diverse patterns of specific phosphorylation events.

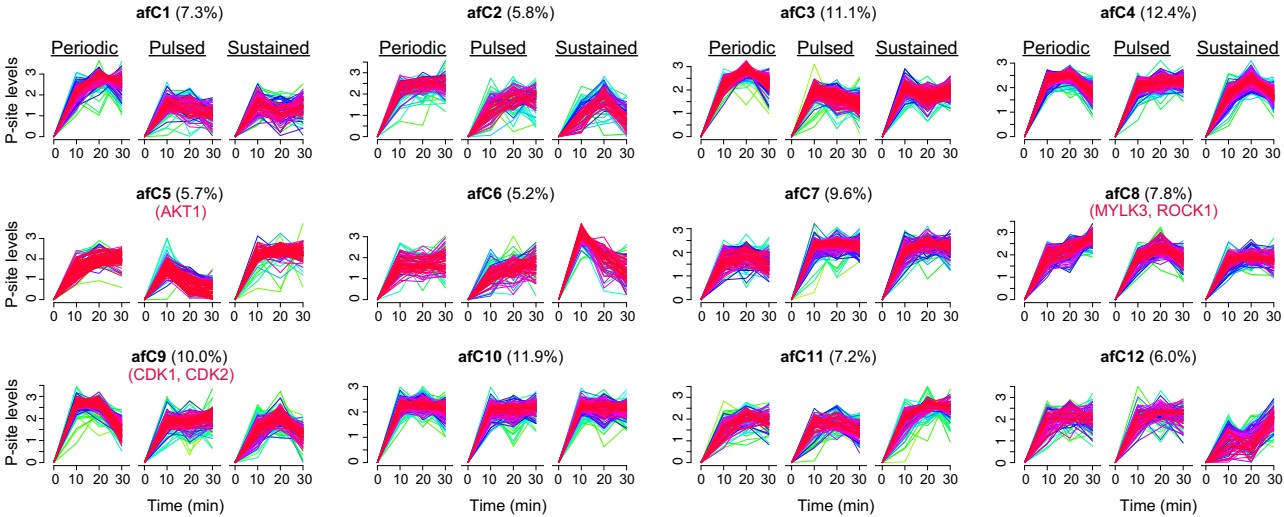

**Fig. 4 | The complex P-site profiles corresponding to the strength, duration, and stability of Akt1 signaling activation.** The FCM analysis categorized 12 clusters of all Akt1 activating P-sites (afC1-afC12). The percentages and the P-site-level enrichment analysis for responsible kinases (if any) are shown for each cluster. The yellow-green-purple-red color palette corresponds to the low to high membership value for each P-site, a similarity score of vectors to each cluster reported by FCM analysis.

## Identifying P-sites commonly and preferably regulated by growth factor and Akt1 signaling

Because pharmacological perturbation experiments might exert off-target effects and genetic approaches have limited temporal control, we considered that Optop-DIA may provide unique advantages in dissecting cell signaling networks activated by key angiogenic factors, such as angiopoietin-1 (Ang1). Ang1 is an EC survival factor that activates the AKT pathway to support cellular functions[59]. As a growth factor, Ang1 plays an important role in vascular development and angiogenesis[60,61]. Thus, we stimulated EA.hy 926 cells with Ang1 for 10, 20, and 30 min and subjected these samples to phosphoproteomic measurements. Previously, Ang1 was shown to exert the antiapoptotic effect through Akt1 phosphorylation in a PI3K-dependent manner in EC cells, which can be blocked by LY294002[62]. Importantly, Ang1 also stimulates the extracellular signal-regulated kinase 1/2 (ERK1/2) pathway in EC cells[60,63,64], a key signaling cascade that crosstalks with the Akt1 pathway[2]. While light-induced Akt1 phosphorylation did not lead to obvious ERK phosphorylation, we found as expected that Ang1 significantly upregulated p-ERK (Supplementary Fig. 8a). Ang1 increased Akt1 pT308 in a time-dependent manner to a level higher than that observed with light condition Su010 but lower than that with Su025, therefore providing a reasonable comparative benchmark (Supplementary Fig. 8b and Fig. 5a).

Using maSigPro, a tool that extracts differential profiles from time-course datasets[65], we identified a total of 2635 P-sites showing distinct temporal profiles between Su010 and Ang1 experiments (Benjamini-Hochberg adjusted $q < 0.05$, $F$-test by maSigPro), 1893 of which (71.8%) also displayed distinct patterns between Su010 and Su025 (Supplementary Fig. 8c). To classify all differential temporal profiles, we referred to the four-group-based portioning offered by maSigPro (mC1-mC4 clusters, Supplementary Fig. 9 and Fig. 5b). Intriguingly, mC1 ($n = 863$ P-sites) and mC2 (1240 P-sites) are two major clusters, respectively, representing P-sites that are either relatively downregulated or upregulated by Ang1. In contrast, mC3 (391 P-sites) and mC4 (141 P-sites) depict differential temporal profiles in which Ang1 stimulation appears in the middle between Su010 and Su025 (Fig. 5b), reflecting common P-site usage in Akt1 and growth factor signaling. In corroboration, Akt1 pT308 was categorized into mC3, while MAPK3 pY204, a classic marker of ERK activation, was classified into mC2. Next, we utilized P-site-specific enrichment analysis again to infer commonly and preferably used kinase substrates (Fig. 5c).

Examples of P-sites with their annotated kinase in OmniPath[55] are shown in Fig. 5d. In this analysis, the mC3 and mC4 P-site lists were combined together (i.e., mC3&4). Many canonical P-sites downstream of Akt1 and growth factors, such as substrates for PRKAA1, mTOR, PRKCA, PRKCD, and RPS6KB1 were enriched in mC3&4, indicating that they are Akt1-associated nodes common to Ang1 and Akt1 signaling (all $P$ values < 0.05). Perhaps even more interestingly, many other kinases and pathways had specific substrates enriched in mC2 only, such as EGF signaling (example P-site: KDM3B_S798), CAMK2A (PLCB3_S537), FGFR1 (BCAR1_Y128), suggesting that these kinases were poorly induced by Akt1 activation alone. Last but not least, in spite of the substantial number of P-sites in mC1, only a few unique kinases were enriched in that cluster, possibly suggesting indirect mechanisms involving phosphatases or uncharacterized kinases.

In summary, based on the specificity of light-controllable Akt1 and phosphoproteomic analysis, Optop-DIA successfully identified synergistic and specific P-sites using Ang1 as a stimulus for EC activation of Akt1 and additional signaling pathways.

## Benchmarking and validating P-sites co-varying with Akt1 phosphorylation

Sites directly phosphorylated by a given kinase generally co-vary with activating phosphorylation sites on that kinase[66,67]. We thus sought to validate P-sites co-varying with p-Akt1 across all 31 experimental conditions in this study as potential Akt1 substrates. We defined our core phosphoproteomic results—112 P-sites on 77 proteins positively correlated (Pearson R > 0.85) with PDK1-phosphorylated Akt1 pT308 across all conditions—as the "OptoCore" list (see Supplementary Data 1). We first compared OptoCore proteins to potential Akt1 interactors in Bioplex[68] and STRING[69]. Surprisingly, the sets of Atk1-associated proteins included in Bioplex (11 proteins) and STRING (10 proteins at high confidence) did not overlap themselves. The 77 corresponding OptoCore proteins had only two proteins overlapping with STRING: NOS3 and FOXO3. The poor overlap among Bioplex, STRING, and OptoCore suggests independence of the Akt1-substrate relationship and stable protein-protein interaction. Next, we compared OptoCore to a series of datasets containing potential Akt1 substrates. These include (a) 342 P-sites listed as established Akt1 substrates in the PhosphoSitePlus database (as of 03/23/2022; 145 of which were detected in our Phos-DIA experiment; hereafter, PSPdb); (b) 1033 P-sites we detected carrying the Akt1 motif R-x-R-x-x-S*/T* (hereafter,

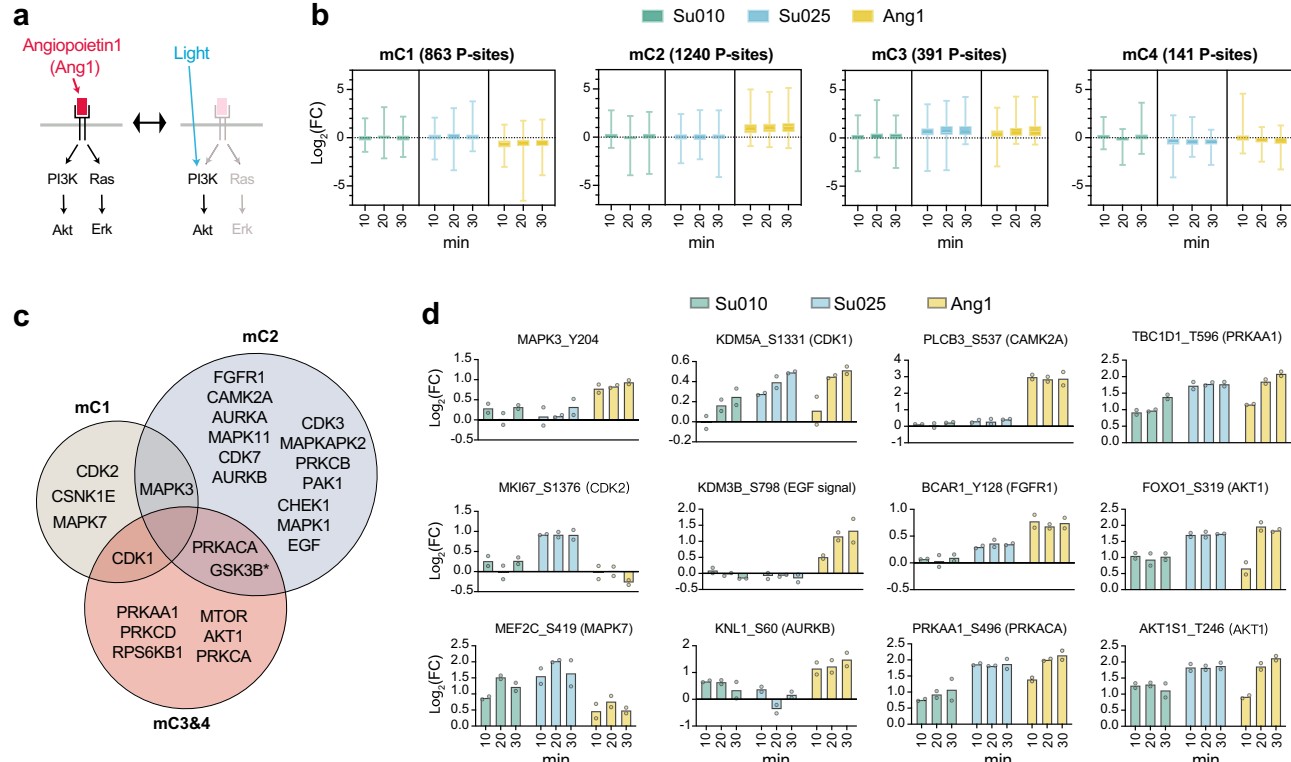

**Fig. 5 | The systemic comparison of light- and Ang1- activated Akt1 downstream signaling. a** Light- and Ang1-activated Akt1 signaling demonstrates an opportunity dissecting the usage of individual P sites for Akt1 and Ang1 signaling. **b** maSigPro analysis classified differential temporal profiles from time-course datasets in four clusters (mC1-mC4), for which the log2 fold-changes Fold changes (FC) values (10/20/30 min compared to 0 min) in each time were visualized in nested boxplots (the number of P-sites for mC1-mC4 clusters are indicated above the boxplots; center

line, median; box limits, upper and lower quantiles; whiskers, min to max values). **c** P-site-specific enrichment analysis for known kinase substrates among mC1-mC4 (all $P$ values < 0.05). In the mC3&4 circle, the asterisk following GSK3B denotes that only GSK3B is enriched from mC4, whereas other kinases are enriched from mC3. Error bar denotes SD. **d** Examples of P-sites with their possible, annotated kinase enriched based on OmniPath and PSPdb ($n = 2$ biologically independent samples, histogram bars denote mean). Source data are provided as a Source data file.

Akt1 motif); (c) 153 P-sites we identified that were effectively down-regulated upon treatment with each one of five Akt1 inhibitors, in a recent study[70] (hereafter, 5 inhibitors); and (d) 257 P-sites with a high Akt1 "kinome score" (Percentile > 0.99, according to a recent study based on synthetic peptide libraries and computational prediction[71] (hereafter, kinome). We found that the Pearson correlation R values to Akt1 pT308 across experiments of these four lists are respectively significantly higher than other P-sites detected by Optop-DIA ($P = 8.79e{-}23$, $1.36e{-}54$, $1.17e{-}36$, and $6.68e{-}23$, respectively, Wilcoxon test; Fig. 6a and Supplementary Fig. 10a–c), indicating a high degree of consistency between OptoCore and the other four lists. Among the 112 P-sites in OptoCore, only 26 were listed in PSPdb (Fig. 6b). Likewise, the P-sites among all the five lists (including OptoCore) overlap poorly with each other (Supplementary Fig. 10a). Reassuringly, analysis of the mutual supportiveness between the five datasets suggested that, while the Akt1 motif list is the least selective, the selectivity of OptoCore is as good as, if not better than, PSPdb (Fig. 6c). PSPdb and OptoCore shared a globally similar distribution frequency of amino acids surrounding the phosphorylated S/T (Supplementary Fig. 10d) and many classic Akt1 substrates[1,2] (Fig. 6d and Supplementary Fig. 10e). It is intriguing to note that majority of OptoCore P-sites tend to have hydrophobic amino acids at the P + 1 position, in agreement with previous reports[72] (Supplementary Fig. 10d). Together, the above comparison underscored the quality and specificity of OptoCore.

To further validate the OptopDIA results, we adopted an orthogonal experimental approach based on an in vitro kinase assay in whole cell lysate and quantitative mass spectrometry[73]. In this assay, the ATP

analog irreversible kinase inhibitor FSBA was applied to the whole cell lysate of non-transduced EA.hy 926 and HUVEC endothelial cells to inactivate all endogenous protein kinases before adding purified Akt1, allowing for identification of direct substrates (Supplementary Fig. 11). Following the kinase reaction, we performed phosphopeptide enrichment and a plexDIA analysis[74] using mTRAQ labeling[75]. We were able to quantify the 6565 and 9468 ratio values for all phosphopeptides (see "Methods") from EA.hy 926 and HUVEC cell lysates respectively. We found the ratios of OptoCore sites were remarkably higher in lysates incubated with Akt1 compared to control lysates ($P = 4.21e{-}20$ and $2.27e{-}30$, Wilcoxon test; Fig. 6e). Moreover, we were able to quantify 19 unique P-sites in the OptoCore list in lysates from both cell lines (Fig. 6f), all of which from EA.hy 926 cells and 18 out of 19 from HUVEC cells were upregulated after Akt1 treatment (2.98-fold and 4.02-fold upregulated on average). Finally, the upregulation of two classic Akt1 substrate P-sites, eNOS pS1177 and GSK3β pS9, as well as two novel sites (in OptoCore but not included in PSPdb), namely NEDD4L pS448 and pNRG1 S330, were verified by immunoblotting using P-site-specific antibodies in both cell lines (Fig. 6g, h).

In summary, as indicated by bioinformatic benchmarking and the additional kinase assay followed by immunoblotting and MS quantification, our OptoCore list and the entire OptoDIA dataset produced a promising resource of novel Akt1 closely-associated P-sites and substrates.

**A collection of Akt1 closely-associated P-sites and substrates**
We finally visualized OptoCore P-sites with a strict AKT motif (R-x-R-x-x-S*/T*) in Fig. 7, with canonical functions marked for well-established

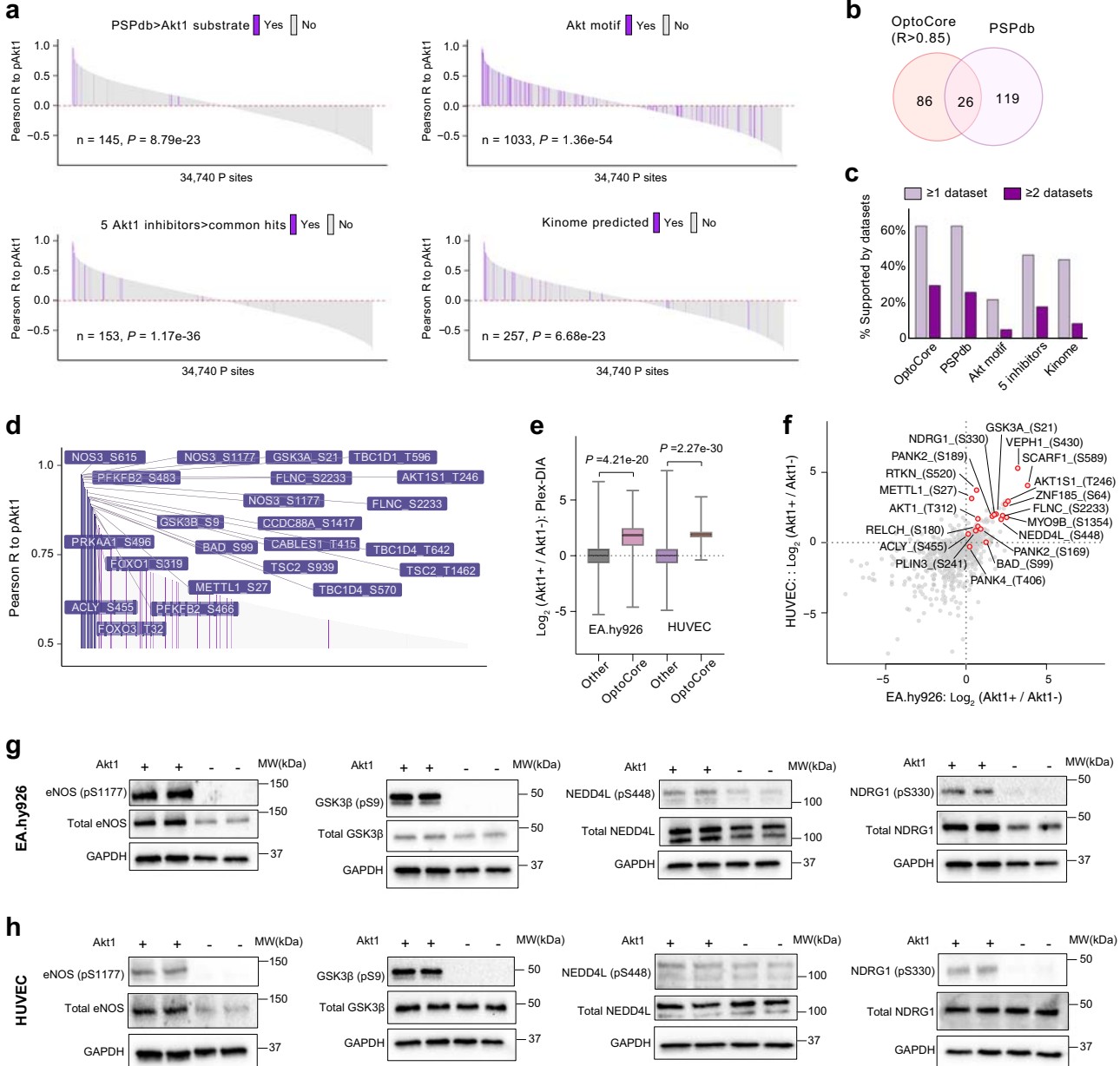

**Fig. 6 | The bioinformatic and experimental validations of P-sites co-varying with pAkt as potential Akt1 substrates. a** Waterfall plots of the *Pearson* correlation R values to Akt1 pT308 for all P-sites across experiments. Those P-sites highlighted in purple are included in each of the four lists containing potential Akt1 substrate P sites (PSPdb, Akt motif, 5 inhibitors, and kinome lists, please see the description in the "Results"). The *P* values are determined by two-sided Wilcox test comparing the *Pearson* correlation R values to Akt1 pT308 between P-sites within and out of each list. **b** Venn diagram for P-sites in OptoCore (i.e., Pearson correlation R values to Akt1 pT308 > 0.85) and PSPdb sites identified. **c** The percentage comparison indicating that for respective P-sites of each of the five lists (OptoCore, PSPdb, and Akt motif, 5 inhibitors, and kinome), how many percent of them are supported by any or any two of the other four lists. **d** Examples of many classic Akt1 substrates shared by PSPdb and OptoCore and their *Pearson* correlation R values to Akt1 pT308. **e** mTRAQ ratios derived from the plexDIA analysis on the in vivo kinase assay based on FSBA for phosphopeptides within and out of OptoCore list. The ratio values of the OptopCore list (*n* = 50 and 61 in EA.hy 926 cells and HUVEC cells) are remarkably higher in the Akt1-added cells compared to the control cells (*P* values, two-sided Wilcoxon test). This is visualized by boxplot (center line, median; box limits, upper and lower quantiles; whiskers, min to max values). **f** The mTRAQ ratio values (Akt1 added/Control) for 19 unique P-sites in OptoCore list quantified in both cell lines. **g**, **h** WB analysis assaying the kinase assay results for two classic Akt1 substrate P-sites (eNOS_S1177 and GSK3β_S9) and two novel P-sites, NEDD4L_S448 and NRG1_S330, both are not included in PhosphositeDB in EA.hy 926 cells (**g**) and HUVEC cells (**h**). For (**g**, **h**), two times of WB experiments were repeated independently with similar results. Source data are provided as a Source data file.

Akt1 downstream sites and a Reactome category associated with Signaling by Akt1 E17K in Cancer. In addition to the sites shown in Fig. 7, due to the reproducibility of DIA-MS and specificity of the optogenetic system, we consider the overall dataset a resource for future Akt1-related studies and systems analysis of cell signaling. We generated an Akt1 Optop-DIA Website (https://yslproteomics.shinyapps.io/AKTPhos/) to navigate this dataset (Supplementary Fig. 12a). This website provides queries at both P-site and protein levels across all optogenetic and Ang1 experiments and data visualization through histograms and clustered heatmaps. Our resource will facilitate generation of novel biological hypotheses related to Akt1. For example, we identified a total of 43 histone phosphorylation events, with surprisingly heterogeneous regulation between H1 and H2A/B (Supplementary Fig. 12b), indicating potential associations between Akt1 activation

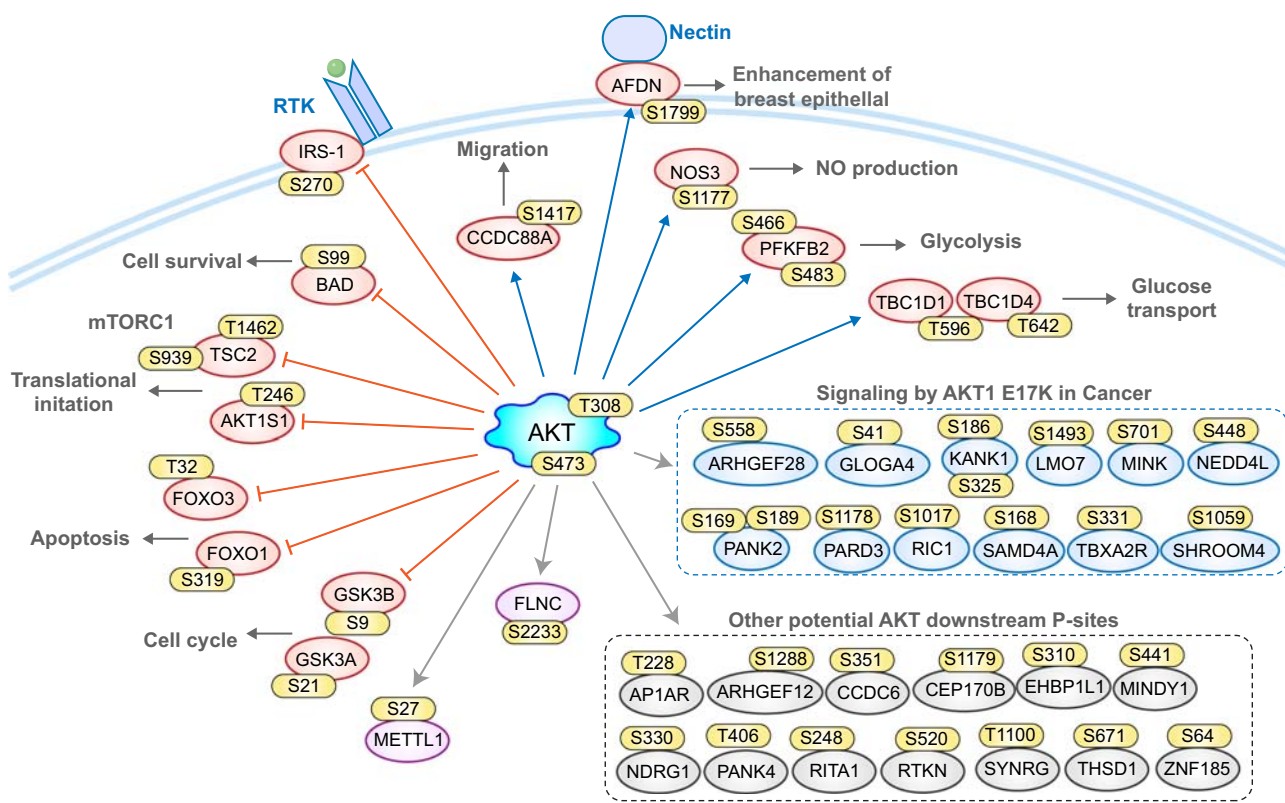

**Fig. 7 | P-sites identified by Optop-DIA with Pearson R > 0.85 to Akt1_T308 and carrying AKT motif.** The orange arrow indicates that the inhibition mechanism well-established, and the blue arrow indicates the activation regulation. The gray arrow indicates unclear inhibition or activation.

and specific histone modifications. Supplementary Data 1 also provides access to the entire dataset, incorporating clustering results and annotations from Figs. 2–5.

## Discussion

Despite more than 30 years of study[76], aspects of AKT signaling have yet to be fully addressed, including how temporal patterns of AKT activation contribute to downstream outcomes. The data we generated in the present study provides substantial insight into how Akt1 processes information to regulate downstream patterns of phosphorylation and cellular functions. Firstly, we found that a ~10-fold increase in the level of p-Akt1 could perturb up to 48% of the total cellular phosphoproteome in less than 30 min. Even though the breadth and complexity of the Akt1 signaling network is well-recognized[2], direct systems-level analysis has been lacking. Our results emphasize the central position of Akt1 as a master node in rapid signaling responses and, broadly, the essentiality of using a systems perspective to understand the signaling network. Secondly, our data highlight the precise temporal control of Akt1 signaling. Utilizing pulsed illumination, we found that, although most P-sites perturbed were correlated with Akt1 signaling intensity, this regulation frequently exhibited a non-linear relationship with intensity. Indeed, direct Akt1 substrates decayed synchronously with p-Akt1, yet their transient phosphorylation rapidly triggered additional dynamic protein phosphorylation events downstream. As suggested by motif and P-site enrichment analyses, after shutting off p-Akt1, general MAPK signaling persists longer than does phosphorylation of direct Akt1 substrates, whereas substrates of CDKs show sustained increases in phosphorylation. Although non-stochastic, sequential waves of cell signaling would be an expected observation, we identified respective P-sites with their specific patterns of decay. Thirdly, the cross-condition FCM analysis revealed that variations in the strength, duration, and stability of Akt1 signaling can lead to diverse sets of downstream P-sites. This

striking complexity might be attributed to prevalent negative feedback regulation in the Akt1 network, which ensures that downregulation of Akt1 signaling is as central to proper function as turning it on[2]. Particular results, such as the stronger upregulation of MYLK3 and ROCK1 substrates under oscillatory pulses compared to constant stimulation, prompt future mechanistic studies. Fourthly, we applied Optop-DIA to dissect P-sites commonly and preferably used in between pathways. The RAS-ERK and PI3K-Akt1 pathways reportedly intersect and compensate for each other[77]. These two pathways can converge to regulate many common effectors, such as FOXO transcription factors[2,77] and mTORC1[78]. Our results recapitulated P-sites commonly used by ERK and Akt1 by classifying them into mC3. Our analysis additionally pointed out P-sites and kinase activities preferably used by the growth factor stimulation of pathway (mC1-2). The above fundamental observations, in essence, sharpen our view of Akt1 regulation and more broadly depict the complexities of temporal cell signaling. Though it did not employ large-scale phosphoproteomics, a recent study compared the effects of insulin stimulation with those of opto-Akt2 in C2C12 skeletal muscle cells[24], further highlighting the utility of optogenetic control in dissecting AKT function.

With Optop-DIA, light can be applied with high precision and minimal invasiveness. The focus on EC allows us to illuminate Akt1-dependent processes, in which Akt2 and Akt3 likely play a minor role[16]. Careful assessment of the optogenetic system ensures judicious selection of phosphoproteomic sampling time points and light intensity ranges. Last but not least, the usage of DIA-MS and P-site localization algorithms[34] is pivotal for confident P-site assignment and reproducible quantification across 31 conditions.

Throughout interpretation of our data, we exploited a series of clustering analyses. However, functional annotations for most P-sites are not available. For instance, although about 91.5% of all the P-sites we measured are included in PSPdb, only 7.9% are annotated with a kinase-substrate relationship in PSPdb, whereas OmniPath annotated

16.4%. On the other hand, protein-level annotation frameworks such as Metascape[47,79], although much more complete, are not useful for phosphoproteomics since most phosphorylation functions are site-specific rather than protein-specific (Supplementary Fig. 7b). For example (Supplementary Fig. 12c), we measured 25 unique P-sites for TSC2 that changed to varying degrees across biological conditions. However, only six of these P-sites carry an AKT motif, with two of the six (TSC2 pS939 and pT1462) showing strong covariation with p-AKT. Therefore, we only performed protein-level analysis for clusters of very small size (e.g., in Fig. 2d and Supplementary Fig. 7c). Considering the above challenges and the substantial P-site-specific differences we identified between distinct illumination regimes, the field will benefit from site-specific PTM annotation databases[80,81] and bioinformatic framework incorporating new phosphoproteomic results[57,82–84]. We believe that both comprehensive P-site-specific annotation and high-quality phosphoproteomic studies are necessary to deeply understand the complexities of cell signaling.

We do note some limitations to the current study, some of which may be worth further exploration. One caveat for any optogenetic system is that it cannot emulate all the steps during natural signaling processes. In particular, the optogenetic tool used in the present study utilized the CRY2-CIBN dimerizer system to recruit Akt1 to PM, facilitating Akt1 phosphorylation by endogenous pathways. Accordingly, Akt1 signaling originates from PIP3-activated, PM-resident Akt1, but not Akt1 that is active on other PIP3- or PI(3,4)P2-containing endomembrane compartments, which reportedly can also support Akt1 signaling[85–87]. We note, however, that key downstream substrates of Akt1 not associated with the cell periphery were detected in our dataset, including the mitochondrial pro-apoptotic protein BAD and FOXO transcription factors. Indeed, the substantial overlap in P-sites between our OptoCore list and other high-quality datasets, as well as our subsequent independent verification experiments, together demonstrate the quality and specificity of the OptopDIA dataset (Supplementary Fig. 10e). While the analysis presented here examined the outcome of short term Akt1 activation, investigation of longer-term activation (i.e., hours or days) might provide further insight into how signaling patterns eventually shape the cellular proteome and phenotype[38]. Also, it is possible that particular kinase-substrate interactions are cell-type dependent, whereas in our study we examined signaling only in EC. Finally, other modifications, such as protein oxidation, may interplay with Akt1 phosphorylation in defining cell signaling outcomes[88]. Like other bottom-up proteomic strategies, DIA profiling on individual phosphorylated peptidoforms can suffer from imperfect PTM localization (especially when discriminating adjacent sites of PTM), incomplete coverage of P-sites via tryptic digestion, and lack analytical power on coordinated functions of multiple PTM sites on the same protein[84]. Other PTMs and PTM crosstalk events in addition to phosphorylation will also be interesting to measure following optogenetic stimulation by combining PTM enrichment and DIA-MS or proteoform-level analysis in future studies. Despite the above limitations, our resource uniquely combined the specificity of optogenetic stimulation and comprehensiveness of Phos-DIA for studying Akt1 signaling. It presents a highly curated, experimentally derived and validated substrate list downstream of Akt1 (direct or indirect), which could serve as a convenient and immediate reference for researchers to assess the Akt1 relevance of their target protein or P-site in the future.

Recently, optogenetic systems have been elaborated to selectively activate Ras[89], beta-catenin[90], Raf/ERK[22], AKT[21], and other pathways with high specificity and temporal resolution[20,22,91]. The latest powerful tools for optogenetic control of ERK and AKT signaling were developed to precisely intervene in reversing neural damage in the central nervous system of live Drosophila larvae[25]. Moreover, optogenetic tools enabling combined temporal and subcellular control of enzyme activity are now available[27]. The optogenetics approach has shown promising therapeutic potential in treating neurological diseases and cancers[92,93] by manipulating the activity of specific cells or proteins in disease models in a highly controlled manner, while proteomics and phosphoproteomics can be used to refine the optogenetic parameters, understand the molecular response to these manipulations, and identify potential drug targets and therapeutic opportunities to improve their safety and efficacy. We therefore believe it is imperative to combine these optogenetic technical advances with cutting-edge phosphoproteomic measurements to systemically elucidate spatiotemporal cell signaling dynamics[26], which remains largely unexplored.

In conclusion, the present study establishes a high-quality resource that expands our understanding of Akt1 function, supporting future Akt1-related studies in many ways. Our Optop-DIA analysis revealed the extraordinary complexity underlying the cellular signaling machinery of Akt1 in orchestrating the levels, oscillatory patterns, and duration of downstream phosphorylation events.

## Methods

### Cell culture and generation of optogenetic Akt1 cell line

The information for all reagents and their resources can be found in Supplementary Table S1. HUVECs (P3) were cultured in EGM2 (Lonza) on dishes coated with 0.1% gelatin. EA.hy 926 cells were cultured in M199 media (Lonza) supplemented with 10% FBS and HAT. The optogenetic constructs, mCherry-CYR2-Akt1 and GFP-CIBN-CAAX, were gifted from the Toomre lab. The lentiviral construct was generated using pLenti CMV Hygro DEST vector in the DB3.1 cell strain. Lentiviral constructs, psPAX2 packaging plasmid, and pMD2G envelop plasmid were mixed in the ratio of 2:1:0.4 in optiMEM (Gibco) and then gently mixed with Lipofectamine 2000 (Thermo Fisher Scientific) also in optiMEM (Gibco). The constructs and Lipofectamine mixture were then added into 60–70% confluent HEK 293T cells overnight. The following day, the media was changed to full (10% FBS) DMEM media. After 24 h, the media was collected and filtered with a 0.45 μm filter for efficacy testing. After 48 h, the media was collected, filtered, aliquoted (2 ml/tube), and stored at −80 °C for future use. To test the efficacy of the lentivirus, 1:1000 polybrene was added and virus used to transduced EA.hy 926 cells. After 72 h, the dish of EA.hy 926 cells was observed under a fluorescence microscope to check GFP and mCherry expression. After efficacy was confirmed, FACS was used to sort cells with strong GFP and mCherry expression levels for the generation of a uniform, stable cell line expressing both mCherry-CYR2-Akt1 and GFP-CIBN-CAAX.

### Optogenetic system for light control

The light system contains LED light (465 nm, 225 Blue LED 14 W, HQRP), a timer (457Z, NSI), a light diffuser (Inventables), a light filter sheet (#65630033, B&H), and a lifting table. Light is controlled by adjusting the lifting table and changing the height of light source. Light intensity was measured by a light meter (model: S170C, THORLABS).

### Live-cell imaging

Spinning disk confocal microscopy was performed with a 60 × 1.45 objective and Volocity software (Improvision). Green laser (488 nm) light was used to induce the interaction of mCherry-CYR2-Akt1 and GFP-CIBN-CAAX. Red laser (561 nm) light was used to visualize the translocation of mCherry-CYR2-Akt1. The green laser exposure time was ~200 ms with 0.2 Hz frequency. The cells were imaged at room temperature for a short time frame (~10 min).

### Cell viability assay

Non-transduced parental EA.hy 926 and HUVEC endothelial cells were exposed to sustained light at 0.25 mW/cm² for 30 or 60 min. At each time point, cells were detached with trypsin and stained with trypan blue, and cell viability was counted using TC20 automated cell counter (Bio-Rad).

## Immunoblotting

To measure protein levels, cells were cultured in Petri dishes, washed twice in cold PBS, and collected in lysis buffer on ice. The lysis buffer contained 50 mM Tris·HCl (pH 7.4), 0.1 mM EDTA, 0.1 mM EGTA, 1% Nonidet P-40, 0.1% sodium deoxycholate, 0.1% SDS, 100 mM NaCl, 10 mM NaF, 1 mM sodium pyrophosphate, 1 mM sodium orthovanadate, 25 mM sodium β-glycerophosphate, 1 mM Pefabloc SC, and 2 mg/mL protease inhibitor mixture (Roche Diagnostics). Protein concentrations were determined using the DC Protein Assay Kit (Bio-Rad). The lysates were mixed with 6x protein buffer (70 ml Tris-HCl, 36 ml glycerol, 10 g SDS, 6 ml 2-methylbutane, 40 mg bromphenol blue in 120 mL of water) and boiled for 5 min. Cell lysates (20–30 μg protein samples) were subjected to SDS-polyacrylamide gel electrophoresis (PAGE) and transferred into 0.45 μm nitrocellulose membranes (Bio-Rad). Membranes were then blocked in 5% bovine serum albumin (BSA) for 1 h at room temperature and incubated with primary antibody at 4 °C overnight. The following day, membranes were washed with TBST (Tris-buffered saline, 0.1% Tween 20) (5 min for 3 times), incubated in secondary antibody at room temperature for 1 h, visualized on a Li-COR Odyssey machine, and analyzed using Image Studio software (Li-COR).

## Nitric oxide (NO) production assay

Nitric oxide (NO) release into the cell culture media was measured with a Nitric Oxide Analyzer (Sievers 270B) according to lab standards[94]. The system measures nitrite, the major product of NO, which is generated when NO is exposed to superoxide anion. Therefore, NO that is produced in the media is able to be measured as nitrite after exposure to iodide and acetic acid at room temperature. Ionomycin from streptomyces conglobatus (Sigma) was added 1 h before collection to serve as a positive control. The media was collected for NO measures, and the cells were collected in protein lysis buffer to measure the protein level. All NO measurements were corrected for total protein (mg) based on whole cell lysates.

## Phosphoproteomics: sample preparation

Lentivirus-infected EA.hy 926 cell lines were cultured in 10-cm dishes in the dark. Once cells reached almost full confluence, the DMEM culture media was changed to phenol red-free media to avoid color influence on the light exposure. Cells were then starved for 5 h and treated with Angiopoietin-1 or light for 0 (control condition), 10, 20, or 30 min. The light conditions included: sustained, periodic, and pulse light patterns at 0.05, 0.1, and 0.25 mW/cm$^2$ intensities, respectively. The light exposure time was 10, 20, or 30 min. The growth factor Angiopoietin-1 (Ang1, R&D Systems) was administered at 400 ng/mL following the vendor's instruction. Two biological replicates were included for all stimulation conditions, as well as the quality control samples. After all treatments, the dish was immediately put on liquid nitrogen for 30 seconds to quench the system effectively, followed by the addition of 400 μL 9 M Urea lysis buffer to lysate the cells on the dish. Cells were then dissociated with cell scrapers and transferred into 2 ml Eppendorf tubes. Samples were stored at −80 °C until use. The 9 M Urea lysis buffer includes 20 mM Hepes (pH 8.0), 9.0 M Urea, 1 mM sodium orthovanadate, 2.5 mM sodium pyrophosphate, 1 mM β-glycerol phosphate. Phosphatase inhibitor (1:100, Halt™, 100x, Thermo) was added freshly to lysis cells every time upon usage. Proteins were digested following lysis as previously described[32,34]. Phosphopeptides were then enriched using High-Select™ Fe-NTA Phosphopeptide Enrichment Kit (Thermo) as we described previously[38,39] and desalted using C18 ultra-micro spin columns (Nest). About 1 μg of enriched phosphopeptide mixture was used for DIA-MS measurement.

## Phosphoproteomics: data-independent mass spectrometry (DIA-MS)

An Orbitrap Fusion Lumos Tribrid mass spectrometer (Thermo Scientific) coupled to a nano-electrospray ion source (NanoFlex, Thermo

Scientific) was used as the liquid chromatography-mass spectrometry (LC-MS) system[95,96] with the data acquisition controlled by Xcalibur (Thermo Scientific) for phosphoproteomic analysis. For a few quality control phosphoproteomic experiments in parental non-optogenetic cells, an Orbitrap Fusion Lumos Eclipse mass spectrometer (Thermo Scientific) was used. Peptide separation was carried out on EASY-nLC 1200 systems (Thermo Scientific, San Jose, CA) using a self-packed analytical PicoFrit column (New Objective, Woburn, MA, USA) (75 μm × 50 cm length) with C18 material of ReproSil-Pur 120A C18-Q 1.9 μm (Dr. Maisch GmbH, Ammerbuch, Germany). Buffer A was composed of 0.1% formic acid in water, and buffer B was composed of 80% acetonitrile containing 0.1% formic acid. A 2-h gradient with buffer B from 5 to 37% at a flow rate of 300 nL/min was conducted for phosphopeptide separation. The DIA-MS method was configured to include one MS1 survey scan and 40 MS2 scans of variable windows as previously described[95,96]. The MS1 scan range was 350–1650 $m/z$, and the MS1 resolution was 120,000 at $m/z$ 200. The MS1 full scan AGC target value was set to be 2.0E5, and the maximum injection time was 100 ms. The MS2 resolution was set to 30 000 at $m/z$ 200, and normalized HCD collision energy was 28%. The MS2 scan range was set to 200–1800 $m/z$. The MS2 AGC was set to be 5.0E5, and the maximum injection time was 50 ms. The default peptide charge state was set to 2. Both MS1 and MS2 spectra were recorded in profile mode.

## Phosphoproteomics: data procession

To analyze the Optop-DIA results, the DirectDIA function (i.e., a spectral-library free method[97]) of Spectronaut software v15 was used[44,98]. The DIA runs were all directly searched against the Swiss-Prot protein database (September 2020, 20,375 entries). The possibilities of Oxidation at methionine, Acetylation at the protein N-terminals, and Phosphorylation at serine/threonine/tyrosine (S/T/Y) were set as variable modifications, whereas Carbamidomethylation at cysteine was set as a fixed modification. Overall, both peptide- and protein-FDR (based on Qvalue) were controlled at 1%, and the data matrix was filtered by Qvalue. The data extraction was performed by Spectronaut with default settings and a Q value cut-off of 1% at both peptide and protein levels. In particular, the PTM localization score was strictly kept at >0.75 to ensure the phosphosites were localized[35], similar to Class I confidence[45,99]. The PTM score of 0 was then used for reporting the total number of identified phosphosites and accepting quantitative values in each sample for all Class-I phosphosites identified at the experiment level; and the number of phosphosites or P-sites were all counted based on the unique phosphopeptidoform level (i.e., the phosphopeptides with multiple modifications were regarded as different P-sites)[38,84]. All the other Spectronaut settings for identification and quantification were kept as default[38]. For each localized P-site, the phosphopeptide precursors with the least missing values among all the samples were taken for relative quantification between samples. The quantitative peak areas for phosphopeptides were log2-transformed for downstream bioinformatic and statistical analysis.

## In vitro Akt1 kinase assay in FSBA-treated cell lysates

The in vitro Akt1 kinase assay based was performed for both non-transduced EA.hy 926 and HUVEC endothelial cells, as previously reported[73]. Cells were lysed in Nonidet P-40 buffer (50 mM Tris•HCl, pH 7.8, 150 mM NaCl, 1% (vol/vol) Nonidet P-40, 1 mM PMSF, and protease inhibitors). Lysate (2 mg/ml) was treated with 2 mM 5′-4-fluorosulphonylbenzoyladenosine (FSBA) solubilized in DMSO and placed at 30 °C for 1 h. The sample was then diluted 1:5 with Nonidet P-40 buffer minus protease inhibitors and desalted using Millipore Amicon ultrafiltration columns with a 10 kDa molecular weight cutoff. Following concentration, the sample was diluted to 4 mg/ml with Nonidet P-40 buffer and diluted 1:2 with 2 × kinase assay buffer (40 mM 3-morpholinopropane-1-sulfonic acid (MOPS), pH 7.2, 50 mM β-glycerophosphate, 10 mM ethylene

glycol tetraacetic acid (EGTA), 2 mM $Na_3VO_4$, 2 mM DTT, 50 mM $MgCl_2$, 400 μM ATP). Recombinant Akt1 was added to a final concentration of 0.5% (wt/wt) total protein. Control (no Akt1 added) and kinase-added samples were incubated at 30 °C for 1.5 h. The samples were then subjected to immunoblotting analysis.

## mTRAQ labeling and PlexDIA mass spectrometry

For both EA.hy 926 and HUVEC endothelial cells, two channels of mTRAQ reagent (ABSciex, Cat # 4374771) were used to chemically label the peptides digested from the protein samples resulting from the control (+140.0949 Da, Δ0) and Akt1 treatment (+148.10916 Da, Δ8) conditions following in vitro Akt1 kinase assay. Per channel, a total of 150 ug purified peptides were used for each channel and labeled by three units of corresponding mTRAQ reagent by following the vendor's instructions. Then, for each of the cell line samples, the labeled peptides were mixed, lyophilized, and cleaned by C18 again for phosphopeptide enrichment following the same protocol above. 2 μg of the mTRAQ-labeled phosphopeptides per cell line for plexDIA mass spectrometry[100], using the identical DIA-MS method and MS platform as above mentioned for label-free analysis.

## plexDIA data analysis

Both DIA-NN (v1.8.1 beta 7)[36] and Spectronaut (v17)[44,98] were used for the mTRAQ ratio extraction. For the present pilot plexDIA analysis on phosphoproteomics using DIA-NN, herein, we followed a workflow modified from the original plexDIA report[100]. First, we relied on the heavy mTRAQ channel to generate the library by allowing only one Phospho (STY) per each fully tryptic peptide. And the "mTRAQ_lib-Gen_Human_fromFASTA" procedure recommended by the original plexDIA report was used for the theoretical spectrum library generation with the additional option specified as "fixed-mod mTRAQ, 148.1091618309, nK". For the following step doing DIA data search and mTRAQ ratio extraction, the searching pipeline "plexDIA_MS2_method" and the above-generated library were used. Herein the additional option was set as "fixed-mod mTRAQ, 140.0949630177, nK" and "channels mTRAQ, 0:0.0, 8: −8.0141988132", because we used the heavy channel (i.e., the condition fo Akt1 added cells) for the library generation. The "Phosphorylation" was selected as a variable modification. Other settings were kept as default.

As for Spectronaut, both DIA and DDA raw data acquired for the mTRAQ samples were used for library generation. In the channels setting, the "mTRAQ-Lys0" and "mTRAQ-Nter0" were set as channel 1, and the "mTRAQ-Lys8" and "mTRAQ-Nter8" were set as channel 2. The "Carbamidomethyl (C)" was set as a fixed modification. "Oxidation (M)" and "Phospho (STY)" were set as variable modifications. The "Use RT as iRT" was used as iRT reference strategy. The "In-Silicon Generate Missing Channels" was enabled to improve library coverage. For the data search, the "Cross-Run normalization" was disabled. The PTM Localization Score was set to >0.75. The "inverted Spike-in" workflow[74] was enabled. Other settings were kept as default.

## Signaling network modeling

Downstream Akt1 signaling networks were contextualized using the R package PHONEMeS[56] (PHOsphorylation NEtworks for Mass Spectrometry). By applying integer linear programming implementation for causal reasoning, PHONEMeS finds a path that connects a set of experiment-specific regulated P-sites with co-regulated kinases in a prior knowledge network (PKN). The PKN contains protein-protein and kinase-P-site interactions downloaded from the OmniPath database[58], resulting in a network with a total of 52,386 edges and 21,886 unique nodes. A set of phosphopeptides included in the clusters fC1-fC4 ($n = 1873$) were used as an input for the algorithm after filtering for unique P-sites present in the PKN ($n = 266$) and removing redundant P-sites identified using, for example both singly and doubly phosphorylated peptides.

An average log2 fold change in the Pu025 experiment (10, 20, 30) relative to untreated control was used as a quantitative value for the analysis. Akt1 was selected as an upregulated kinase input. The PKN was pruned by removing all nodes that were not connected 50 steps up- or downstream with the selected P-sites and Akt1. Protein activity networks were constructed by removing the kinase-P-sites interactions from the network. To focus on downstream targets of Akt1 signaling, the final subnetwork was extracted, containing all edges and nodes 3 steps downstream of Akt1.

## Kinase-substrate prediction using PhosR

PhosR R package[57] was used to predict kinase-substrate relationships for unique P-sites in clusters fC1-fC4 by combining the information about dynamic profiles of canonical substrates of kinases across all conditions across the experiment (i.e., not only Pe025) and consensus kinase motif scoring. Predicted kinase-substrate associations were filtered based on a predicted score cutoff of 0.5, and then the annotated P-site matrix was subjected to an enrichment analysis using Fisher's exact test in Perseus v1.6.14.0[101] to identify substrates of specific kinases overrepresented in clusters fC1-fC4. In case of a significant over-representation of a kinase in multiple clusters, the most representative cluster was selected based on the lowest $p$-value. The results of the enrichment analysis were used as a basis for color mapping for the Akt1 signaling subnetworks identified by PHONEMeS.

## Other bioinformatics and statistical analysis

The XIC graphs were reported by Spectronaut v15 and modified in Adobe Illustrator. The motif analysis was conducted with motifeR[102]. The extraction of phosphopeptide containing the general motif (R-x-x-S*/T*) or the strict AKT motif (R-x-R-x-x-S*/T*) was performed by the extract function in motifeR[102]. Pearson coefficients were calculated using cor() in R. The heatmap and cluster analysis were performed using the R package "pheatmap" or by Cluster 3.0 followed by TreeView-based visualization. Principal component analysis (PCA) was performed based on R function prcomp(). Fuzzy c-means clustering (FCM) analysis was performed by the package 'Mfuzz' (version 2.46.0) with the following settings: M = 1.25, Normalization to time point 1. The colors of lines in Fig. 3b and Fig. 4 denote the membership values after FCM clustering i.e., a similarity score of vectors to each cluster[103]. The yellow-green-purple-red color palette corresponds to the low to high membership value for each P-site. The protein-level biological annotation and enrichment analysis were performed by using Metascape [https://metascape.org/][47]. The enrichment $P$ values reported by Metascape are described in their website [https://metascape.org/blog/?p=122]. The P-site-level annotation and enrichment analysis against kinase-P-site relationships and regulatory processes were conducted using the annotation files from Phosphosite database (PSPdb)[54] and OmniPath resources[55] and then the Fisher's exact test function provided by Perseus[104]. The Enrichment factor was set to be >1 and Functional annotations with at least four P-sites were accepted before $P$-value filtering ($P < 0.05$). The differential analysis of P-site profiles between light intensity 010 and 025 groups and Ang1 stimulation were performed using the R package maSigPro[105], with the R-squared of the regression model cutoff of 0.8 (rsq = 0.8) and the number of clusters of 4 (K = 4). The linear ANOVA analysis (Fig. 2) was performed by Pomelo2 (http://pomelo2.iib.uam.es/)[46], with the option "Anova, linear models (limma)". In this ANOVA, additional co-variables were enabled for a more complete analysis accounting for potential interactions between Intensity, Pattern, and Duration traits. This means that one of Intensity, Pattern, and Duration variables was kept as the major factor, whereas the other two as covariates, respectively. GraphPad Prism (v9) was used to generate the histograms for visualizing the different P-sites among Akt1 activation conditions. Figure 1f was created with BioRender.com.

## Reporting summary

Further information on research design is available in the Nature Portfolio Reporting Summary linked to this article.

## Data availability

The mass spectrometry-based Optop-DIA datasets, the validation MS datasets, and processed results in this study have been deposited to the ProteomeXchange Consortium via the PRIDE partner repository[106] under accession code PXD034957. Source data are provided with this paper.

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

## Acknowledgements

mCherry-CYR2-Akt1 and GFP-CIBN-CAAX constructs were gifts from the Toomre lab and were generated by Dr. Yingke Xu. The negative control construct, mCherry-CYR2, was a gift from the De Camilli lab. We thank Dr. Lorena Benedetti for the assistance in the live-cell imaging experiments and providing guidance and advices. We thank Dr. Mark A. Lemmon for helpful discussions. Y.L. thanks the support from the National Institute of General Medical Sciences (NIGMS), National Institutes of Health (NIH) through Grant R01GM137031 to Y.L., as well as a Pilot Grant from Yale Cancer Center and a Career Enhancement Program Grant from the Yale SPORE in Lung Cancer (1P50CA196530). This work was also supported by NIH grant R35HL139945, R01DK125492, P01 HL1070205 to W.C.S.

## Author contributions

W.Z., W.C.S., and Y.L. designed experiments. W.Z. performed all the initial biochemical, imaging, and cell-line experiments for Optop-DIA. Z.H. and B.T. performed additional biochemical and cell-line experiments for validating experiments. W.L. performed the phosphoproteomics and mass spectrometry measurements. W.L. and Y.D. performed the mTRAQ labeling and data analysis. W.L., WZ, S.W., B.S., U.P., W.C.S., B.E.T., and Y.L. analyzed, interpreted, and visualized the data. S.W. built up the AKT website structure. Y.L. lead the writing of the manuscript. Y.L. and W.Z. wrote up the final manuscript with the inputs from other authors. W.C.S. and Y.L. supervised and supported the study. All authors contributed to the review of the manuscript.

## Competing interests

W.C.S. is cofounder of CavtheRx, a member of the scientific advisory board of Alucent and Antibe Therapeutics, and a Senior Vice President and Chief Scientific Officer for the Internal Medicine Research Unit at Pfizer. Other authors declare no competing interests.
