## [Peer Review File · Nature Communications]

Reviewers' comments:

Reviewer #1 (Remarks to the Author):

Zhou et al systemically demonstrated Akt activated/associated phosphorylation profiles upon different Akt stimulation intensity, pattern, and duration. They used an elegant optogenetics approach to stimulate Akt activity, followed by characterization of protein phosphorylation using a novel data-independent acquisition (DIA)-based phosphoproteomics. This study was well-designed, and experiments were carefully carried out combining with innovated informatics pipeline to interpret the data. The proteomics data presents high reproducibility and relatively consistent observations across different biological conditions. The isogenic endothelial cell system provides a clean background to study phosphorylation patterns upon Akt1 activation; therefore, the results have significant impact to understand Akt downstream signaling and provide a useful resource (in ShinnyApp) for the entire community. This study has greatly extended the current understanding of Akt1 signaling in a temporal manner, and the results, underlining the fundamental cellular mechanism of kinase-substrate interactions, should benefit a wide range of readership to the Nature Communications. Nonetheless, I believe a minor revision of this manuscript before acceptance is required. I have some comments and questions to be addressed by the authors.

Major concerns and comments:

1. My major concern of this manuscript is that the authors failed to address/discuss the specificity of their optogenetic experiment: a) whether the light-based kinase activation is only turn on the Akt1 kinase only but no other downstream kinases (or Akt substrates); b) whether the blue light (wavelength 465nm) may ubiquitously affect normal cellular function, or any side effects are expecting. This is the foundation of this manuscript to claim any Akt-dependent downstream signaling.
2. The authors didn't address the total protein abundance large-scale proteomics data despite a few of western blot demonstrating the pan-Akt protein level didn't change much. The total protein abundance should be used to normalize the P-sites, or an additional figure to show the differential expressed protein level of all identified P-sites upon light-treatment in a intensity/time-dependent manner is needed.
3. In Fig 6, the authors claimed 103 P-sites are Akt substrate or associated with Akt activation, interestingly, a vast majority of these (81 out of 103) have not been reported as Akt substrates, which indicating potential false positives being reported here. Since most previous kinase-substrate interactions were performed using co-immunoprecipitation and/or proximity labeling-proteomics, the authors please address why their rules can be an alternative approach for substrate identification. A cross-study of current datasets with previous Akt PPIs should be performed.

Minor concerns and comments:

1. This study only addresses the phosphorylation can be activated by the photoreceptive constructs, the authors should discuss the potential cross talks between phosphorylation and other post-translational modifications, such acetylation and ubiquitination upon light-activation.
2. In Figure 1F, it is not quite clear about the phospho-enrichment efficiency, please elaborate, and provide additional figures/tables as necessary.
3. It is not quite clear how many biological/technical replicates per group were acquired in the experiment, some of the error bars (e.g., Pe025 of AKT_T308 in figure 1G) are not trivial. The inter-sample CV% of each group should be provided in the supplemental material.
4. Can authors discuss more potential applications of optogenetics-proteomics approach in biomedical research.

Reviewer #2 (Remarks to the Author):

The authors described a novel aspect of Akt signaling with combined use of an optogenetic system to activate Akt and quantitative phosphoproteomics techniques. Overall, the experimental design to decipher the temporal aspect of Akt signaling is sound and the obtained result provides potential candidates for novel Akt-mediated phosphosites and proteins. In addition, a Web application of Opto-DIA results makes public access to the authors' findings and contributes to the phosphosites research in the field of Akt signaling research. However, there are no data with other approaches to support their findings by the phosphoproteome analysis, which is critically important to validate the results. In addition, some of the results displayed in the figures are not easy to be interpreted and need to be clarified more. This reviewer felt the authors need to address the following concerns;

Major concerns

1. Representation of data in Figures 2, 3, 4, and 5 is not easy to be interpreted and is not fully explained in the legend. Specifically, please consider the revision of the following figures.

Figure 2D: Can the authors provide scales for color coding of the heatmap? In addition, if there are no related GO terms for hC3, I think the authors can leave the section blank instead of listing detected phosphosites because they are not a GO term therefore somewhat misleading.

Figure 3B, Figure 4: The color coding for each line is unclear. What is the difference between blue and red? In addition, can the authors provide representative behavior for each cluster such as the average profile?

Figure 5B: Labels on the horizontal axis lacks information on the condition for what the data correspond to (is it time?). In addition, is it difficult to combine the result of two replicates for each condition? Also, please consider adjusting the scale for the vertical axis to a constant value for all the clusters.

Figure 5C: Please adjust the scale for each cluster. For instance, mC1 corresponds to phosphosites that decreased in phosphorylation in Ang stimulation and mC2 corresponds to phosphosites that increased in phosphorylation by Ang stimulation, but visually cannot be discriminated. In addition, what does the dotted line represents in the figure?

2. For the FCM analysis in Figure3 and 4, I could not follow why the authors initially filtered phosphosites that were phosphorylated by the Pe025 condition. What would be the difference if filtered by Su025 and/or without filtering? The authors should explain what this filtration process is intended to be in the text. In addition, although I agree that phosphorylation for phosphosites in fC2 delays that in fC1, I think it is too exaggerated to call it a “waves of a signaling cascade” as there is no evidence of causative relationships in responsible kinases for fC1 and fC2 (in other words, the findings by FCM analysis only show that phosphorylation at phosphosites in fC1 and fC2 are timely distinct events). If the authors claim it is a cascade event, they need to clarify the causative relationships or activation time difference of at least one or two potential responsible kinases for those clusters.

3. One of the key findings in this study is the identification of novel Akt-associated phosphosites and proteins. If the authors could verify some of the newly discovered proteins with other approaches (i.e. biochemical), that would be strong supporting data for what the authors claim.

Miner concerns

1. Although the authors selected light illumination conditions and duration to avoid cell toxicity, the effect of light on the phosphoproteome is elusive. Can the authors provide information on the influence of light stimulation on phosphoproteome?

2. The interesting finding in Figure3 is that phosphosites for fC3 are enriched in the MAPK phosphorylation motif. Because the authors state that one of the kinases of the MAPK, ERK, is not activated by optogenetic Akt activation, what could be the responsible kinase for fC3? Can the authors provide data to support their findings, such as presumable activation of ERK, p38, and Jun from their phosphoproteomics measurements?

3. In Figure 3D, can the authors also provide “%with Akt motif” for entire detected P-sites to represent enrichment of Akt motif in fC1?

4. Wherever possible, please consider indicating the specific phosphosites in the text and figures because identification of phosphosites is one of the key findings of this research. For instance, in Figures 1D and E, it would be beneficial to present specific phosphosites of eNOS and GSK3 β .

5. For the enrichment analysis, the authors use motif enrichment for some cases and annotated kinase enrichment analysis for other cases. What are the criteria for selecting the type of enrichment analysis? Could the authors perform both types of enrichment analysis?

6. In Figure 5D, what does the asterisk for “GSK3B*” mean?

7. In L312, the authors describe mC3 & mC4 as “cross-talking nodes”, however, because they are the phosphosites both regulated by optoAkt and Ang, I think they are “Akt-regulated nodes”. I’m curious why the authors stated they are cross-talking.

8. In Figure6, the rules for the search of Akt substrate phosphosites are interesting. Can the authors provide information on the number of phosphosites that matched each rule, and also whether there was any overlap of phosphosites among phosphosites that matches each rule?

9. Several typos are found in the text.

L157: “p-Akt T308” should be “Akt p-T308”.

L304: “Su020” should be “Su025”.

Reviewer #3 (Remarks to the Author):

Summary

The authors of this manuscript present a new workflow for the analysis of Akt signaling in endothelial cells (EC) in which they combine a previously reported optogenetic system (ref #30) for controlling Akt translocation to the plasma membrane with data-independent acquisition mass spectrometry (DIA-MS) (ref #18). As previously reported, the authors demonstrate that the optogenetic system works robustly to permit the light-dependent, reversible translocation of Akt to the plasma membrane. The authors achieve impressive technical results in their DIA-MS pipeline regarding coverage of the phospho-proteome, and have generated a wealth of data regarding the changes in phospho-site abundance following light-stimulated Akt translocation. Using hierarchical clustering analysis and principal component analysis, the authors classify phospho-sites that are upregulated into different groups according to the temporal signatures of the phospho-site abundance. In general, the work is technically well performed, but the insights and conclusions are somewhat limited. As a data resource, the work may (if revised adequately) represent a useful cross-validation tool for published and future studies, while the pipeline has the potential to compare Akt signaling in different cell lines under different stimulation regimes.

Major comments

Introduction / experimental design

The authors do not cite recent and accumulating evidence that Akt is dependent on PIP3 or PI(3,4)P2 for its activity, not just on its phosphorylation by PDK1 and mTORC2. This is important, since the authors cite their own data on page 4 that endogenous PI3K activation is required for light-stimulated Akt activation.

The authors do note in their discussion that their reporter is limited to Akt signaling at the PM, whereas numerous reports have indicated Akt signaling on endomembrane compartments. However, some aspects of their reporting would benefit from additional clarification. Specifically:

(a) Their reporter is not light-activated, as they claim/imply. They use light to control the subcellular localization of Akt, not its activation. As they show themselves, and which is supported by numerous reports, Akt activation depends on PI3K and the lipid second messenger PIP3. As such, their reporter reports on the endogenous pathways that activate PI3K, albeit amplified by increasing the residence time of Akt at the PM. Essentially, the authors have designed a reporter that can interrogate the activity of Akt1 at the PM, since they anchor it there with a prenylated lipid anchor. This means that the signaling pathways triggered downstream of Akt are originating from PIP3-activated, PM-resident Akt, and not Akt that is active on any PIP3- or PI(3,4)P2-containing endomembrane in the cell. The authors should clarify this in their discussion (pg 14 ln 412-416 should be expanded with additional citations). Given the context in which Akt is activated (i.e. by endogenous PIP3), it is questionable what useful information is provided by the differing light stimulation regimes, which induce a non-physiological, reversible translocation of Akt from the cytosol to the PM. What the authors are, in fact reporting on, is a convolution of endogenous PI3K signaling with an artificial, light-dependent recruitment that serves to transiently increase Akt residence times at the PM, during which Akt can be activated more readily.

(b) The recruitment of Akt to the PM using prenylated GFP is an artificial, perturbative mechanism of recruitment and the authors should acknowledge that this has the potential to distort the signaling pathways activated by this lipid-anchored Akt.

Results

The authors use motif enrichment analysis to identify Akt substrates based on a 'strict' Akt consensus motif and a 'loose' Akt consensus motif. The 'strict' motif, however, is simply an AGC kinase consensus motif, of which there are 63 in the human proteome. It is also erroneously cited as being R-X-R-X-X-S/T

when in fact the minimal Akt recognition motif was reported to contain a bulky hydrophobic in the P+1 position. Using this motif as a filter will undoubtedly lead to many false positives. The 'loose' motif, however, is even more problematic, since Akt is known to depend on Arg in the P-5 position of its substrates (it forms a specific salt bridge with Glu279 in the kinase domain, Yang et al., NSMB 2002) and its reporting as an Akt consensus motif is based on questionable cell biological data. The authors of the review article cited by the authors (Manning and Toker, Cell 2007) considered (as I do), on balance of all the available data, an R-X-R-X-X-S/T motif to be an essential feature of bona fide Akt substrates.

Pg 5, ln 139-140. The authors report the phosphoproteomic analysis of cells in duplicates. It is not clear from this or the methods section whether these 'duplicate' samples were true biological replicates or technical replicates of the same cells. This is important since, if they are technical replicates, the authors are reporting only one biological experiment on one cell line. Moreover, since the reporter actually reports on endogenous Akt activation by PIP3 (see pg 4, ln 108-111), considerable heterogeneity is likely to exist between individual cell populations grown under the same conditions, but at different times and in (potentially) different batches of growth medium. Whilst it is a significant amount of work, at least two biological replicates should be analyzed to get a sense of the reproducibility of the signaling networks activated downstream of Akt in endothelial cells.

Minor comments

Pg. 7. P-sites classified in hC1 that persist for >30 min after switching off the light (and p-Akt with it) are not likely to be Akt substrates, but the consequences of signaling pathways upregulated by transient Akt activation. The authors imply that they could be Akt substrates in the nucleus, but this would necessitate them being phosphorylated at the PM, where the authors' reporter is localized. The authors should clarify that these 'substrates' are unlikely to be direct Akt substrates.

The manuscript generally contains a lot of unnecessary, inflationary and emotional verbiage that is not scientific and should be avoided. Examples include, but are not restricted to:

Pg. 2, ln 64-65: "how cells decode the same signaling input from Akt and that result in distinctive downstream signaling defines a quintessential problem in modern molecular biology that remains largely unknown."

Pg. 3, ln 77: "Conceivably, it would be straightforward and exciting to..."

Pg. 5, ln 142: "Using our high-performance phosphoproteomics-DIA..."

Pg. 11, ln 325 "Due to the favorable reproducibility..."

Pg. 9, ln 277: "...representing an uncharted complexity in Akt signaling."

Pg. 8, ln 234: "...remarkably enriched in..."

Pg. 9, ln 267: "afC6 is fascinating as well, because..."

Pg. 7, ln 214: "...exhibits intriguing behaviors."

The authors use the words 'dynamics' and 'cross-talk' repeatedly in a very unspecific manner which is not helpful to the reader.

The manuscript contains a large number of typos, spelling, punctuation and grammatical errors.

Figure 3B and Figure 4. There is no legend provided to explain the color code used in these panels. Presumably, each line represents an individual phospho-site, but the meaning of the different colors is unclear.

Supplementary Figure 1 legend. The legend to panel B is non-sensical and needs considerable revision.

Supplementary Figure 5C. There are more rows to the table of data presented than there are GO terms to annotate them. The missing terms should be added and the terms aligned properly with the table rows.

Reviewer #1 (Remarks to the Author):

Zhou et al systemically demonstrated Akt activated/associated phosphorylation profiles upon different Akt stimulation intensity, pattern, and duration. They used an elegant optogenetics approach to stimulate Akt activity, followed by characterization of protein phosphorylation using a novel data-independent acquisition (DIA)-based phosphoproteomics. This study was well-designed, and experiments were carefully carried out combining with innovated informatics pipeline to interpret the data. The proteomics data presents high reproducibility and relatively consistent observations across different biological conditions. The isogenic endothelial cell system provides a clean background to study phosphorylation patterns upon Akt1 activation; therefore, the results have significant impact to understand Akt downstream signaling and provide a useful resource (in ShinyApp) for the entire community. This study has greatly extended the current understanding of Akt1 signaling in a temporal manner, and the results, underlining the fundamental cellular mechanism of kinase-substrate interactions, should benefit a wide range of readership to the Nature Communications. Nonetheless, I believe a minor revision of this manuscript before acceptance is required. I have some comments and questions to be addressed by the authors.

Author reply: We thank the reviewer for the overall positive comments.

Major concerns and comments:

1. My major concern of this manuscript is that the authors failed to address/discuss the specificity of their optogenetic experiment: a) whether the light-based kinase activation is only turn on the Akt1 kinase only but no other downstream kinases (or Akt substrates); b) whether the blue light (wavelength 465nm) may ubiquitously affect normal cellular function, or any side effects are expecting. This is the foundation of this manuscript to claim any Akt-dependent downstream signaling.

Author reply: In the revision we have performed new phosphoproteomic and cell viability experiments which fully addressed these concerns (see below). We also improved the relevant descriptions.

Firstly, we provide more background on the system. The optogenetic system has been well-tested and published ¹. This system based on the *Arabidopsis* basic helix-loop-helix transcription factor CIB1 and cryptochrome 2 (CRY2), proteins that do not exist in human and mammalian cells. Blue-light illumination induces the heterodimerization of CRY2 with the N-terminus of CIB1 (CIBN) via a rapid reaction that does not require exogenous cofactors ². As compared to chemical inducers, optogenetic systems were widely acknowledged due to its specificity at different levels and less side effects ^{3,4}. We now add these descriptions in **Page 3 Line 82** and **Page 3 Line 77**.

The concerns over phototoxicity come from experiments where blue light irradiation is used for imaging of fluorescent proteins, where much higher intensity light is required for the purposes of sensitivity. Very importantly, we now clarified that the intensity dosage of blue light applied in our phosphoproteomics investigation was very low. In a recent report, Alghamdi *et al.* assessed the phototoxicity of blue light in the mammalian cell line PC3 ⁵. In their experiment, 14 mW/cm² of blue light was used as the “minimally invasive excitation”, whereas 0.2 mW/cm² (close to our highest intensity which is 0.25 mW/cm²) was used as a *control* condition. The authors assessed the blue light toxic effect at 14 ~662 mW/cm², which are 56 to 2648-fold greater than the highest intensity we used in our experiments. The authors found that at 14 mW/cm² (i.e., 56-fold over 0.25 mW/cm²) the cell motility difference is not significant over 24 hours and that a small hormetic trend was only visible at or above 112 mW/cm² (i.e., 448 folds of 0.25 mW/cm²). Consistent to Alghamdi et al, using the same optogenetic system of Akt activation and dead cell staining, Ong et al found that the level of blue light of 0.2 mW/cm² did not cause long-term phototoxicity over 24 hours ⁶. Therefore, the 0.25 mW/cm² for (only) 30 mins of lighting used in our phosphoproteomic investigation present an extremely low intensity and duration to “ubiquitously affect normal cellular function”. Please see **Page 5 Line 139** for the relevant clarification.

Secondly, to fully eliminate the reviewer’s concern in the cell line we used, we have performed two new experiments during the revision.

-We stimulated the normal EA.hy 926 a nd HUVEC endothelial cells (i.e., ***non-transduced cells***) with the blue light of our highest intensity and duration condition (i.e., 0.25 mW/cm² for 30 mins) and preformed new phosphoproteomic DIA-MS measurements. With three biological replicates, we could not detect any global variance higher than the variance between biological replicates, i.e., cells under blue light stimulation did not cluster together in the heatmap (new **Figure S3A**; pasted below, left panel), which is in stark contrast to Figure 2A. As expected, no abundance difference was observed for individual established Akt substrates (such as those P-sites in Figure 1G). This experiment thus confirmed that the blue light condition we used was sufficiently low that it did not detectably perturb the phosphoproteome.

-We also performed cell viability assays on these normal cells above. This means, after the cells being treated with light at 0.25 mW/cm² for 30 minutes, cells were released by trypsin and stained with trypan blue and cell viability was countered using

a TC20 automated cell counter. Indeed, from the percentage of trypan blue-negative cells, no substantial change in cell viability was observed (new **Figure S3B**; see below, Right panel). Please see the corresponding Results text at **Page 5 Line 140**.

Figure S3A-B

2. The authors didn't address the total protein abundance large-scale proteomics data despite a few of western blot demonstrating the pan-Akt protein level didn't change much. The total protein abundance should be used to normalize the P-sites, or an additional figure to show the differential expressed protein level of all identified P-sites upon light-treatment in a intensity/time-dependent manner is needed.

Author reply: We appreciate this comment. Actually, our time-course focusing on the early phosphorylation regulation was particularly designed to avoid such a concern. This is because the protein expression and abundance regulation following an environmental perturbation generally takes more than 30 mins to be detectable by quantitative proteomics ⁷. Within 30 mins, protein post-translational modifications (PTMs) and protein-protein interactions (PPIs) are mostly playing the role. This is reflected by many published pulsed stable isotope labeling by amino acids in cell culture (pSILAC) datasets in human cell lines, in which heavy SILAC labelled proteins (i.e., newly synthesized proteins) are overall barely detectable within 0.5-1 hour without enrichment and much less abundant than the light versions ⁸, indicating that protein turnover is minimal on this timescale. Therefore short perturbations are unlikely to cause general abundance changes that could be detected by current mass spectrometry in a given label-free experiment.

To provide further evidence that our phosphoproteomics experiments are not confounded by changes in protein abundance, we performed a new proteomic experiment to fully address this concern. Herein, we have measured directly the proteome (not phosphoproteome) of transduced EA.hy 926 cells (the same cells used in Optop-DIA) following blue light treatment using our highest intensity and duration condition (i.e., 0.25 mW/cm² for 30 mins). Indeed, while the phosphoproteome underwent large regulation as we shown in the article, the global proteome did not show any reproducible variance between the control and experimental conditions (heatmap and scatterplot below) in such a short time (new **Figure S3C-D**, see below). We therefore suggest that the normalization requested by the reviewer is not necessary in this case.

Figure S3C-D

Please also see these new Results at **Page 5 Line 153** and **Page 5 Line 157**.

3. In Fig 6, the authors claimed 103 P-sites are Akt substrate or associated with Akt activation, interestingly, a vast majority of these (81 out of 103) have not been reported as Akt substrates, which indicating potential false positives being reported here. Since most previous kinase-substrate interactions were performed using co-immunoprecipitation and/or proximity labeling-proteomics, the authors please address why their rules can be an alternative approach for substrate identification. A cross-study of current datasets with previous Akt PPIs should be performed.

Author reply: We thank the reviewer for this comment and suggestion! We believe that we have greatly improved both bioinformatic and experimental validation for the newly discovered Akt substrate or associated P-sites in the revised manuscript.

To facilitate presentation, we define our core phosphoproteomic data (i.e., those 112 P-sites showing $R > 0.85$ quantitative correlation to Akt p-T308) as the **OptoCore** list. These 112 P-sites are derived from 77 proteins (See Columns A-C and AU of **Table S1**).

Firstly, following this suggestion, we compared *OptoCore* with the queried the protein-protein interaction (PPI) datasets of Akt1 interactors in both Bioplex⁹ and STRING databases¹⁰. Interestingly, Bioplex and STRING (high confidence, STRING score > 0.700) reported 11 and 10 Akt1 interacting proteins respectively; but they do *not* overlap at all. The 77 protein IDs of our *OptoCore* had only two proteins overlapping to STRING, which are NOS3 and FOXO3 (**Figure R1** on the right). The small overlap among all three lists (Bioplex, STRING, and *OptoCore*) therefore is strongly suggestive of the dynamic/transient interactions of Akt1 and the independency of Akt1-substrate relationship on stable protein-protein interaction. This is further substantiated by the absence of previously established AKT substrates from the Bioplex list. We add this description in the Results section (**Page 12 Line 357**).

Figure R1

Secondly, in the revision, we have comprehensively compared our *OptoCore* list (n=112 P-sites) to a series of state-of-the-art datasets to interrogate the specificity and quality of our dataset. These four lists are:

- The 342 P-sites listed as Akt1 substrates in Phosphosite database (We identified 145 P-sites. Hereafter, **PsPdb**).
- The 1033 P-sites we identified carrying the Akt motif R-x-R-x-x-S*/T* (Hereafter, **Akt motif**)
- The 153 P-sites we identified that were effectively downregulated upon treatment with all five Akt inhibitors, according to a recent study from the Kuster lab¹¹ (Hereafter, **5 Inhibitors**)
- The 257 P-sites which have an Akt “kinome score” with the percentile > 0.99 (top 1% scoring sites), according to a recent atlas study published in Nature using synthetic peptide libraries and computational prediction¹² (Hereafter, **Kinome**)

We summarized the comparison result in new **Figure 6**. Please see the Results at **Page 12 Line 363**. Basically, our *OptoCore* list, despite its smaller proteome coverage as a single study, showed a high consistency to all of the four lists, with a *selectivity* comparable to, if not better than, PsPdb, both of which are better than the other three lists (**Figure 6C**). This means, although a large fraction of P-sites in our *OptoCore* list was not reported by PsPdb, they are high-quality putative Akt substrate P-sites and thus substantiate the value of *OptoCore*.

Finally, we validated some P-sites in *OptoCore* via both western blots and a new mass spectrometry experiment of PlexDIA (Please refer to the reply to the first point of Reviewer 2).

Minor concerns and comments:

1. This study only addresses the phosphorylation can be activated by the photoreceptive constructs, the authors should discuss the potential cross talks between phosphorylation and other post-translational modifications, such acetylation and ubiquitination upon light-activation.

Author reply: Following this interesting suggestion, we have tried to perform an additional modification search for peptide ubiquitination in our phosphoproteomic dataset using the directDIA function of Spectronaut. However, only tens of peptides were confidently identified and checked to carry both ubiquitination and phosphorylation. Manual inspection suggested that most peptides with both ubiquitination and phosphorylation tended to follow the regulation trend of the phosphorylated form (i.e., the P-sites). Please see an example below (**Figure R2**), which indicates that ubiquitination of the phosphorylated peptidiform might

happen later than 30 mins. Because current acetylation-proteomic analyses normally yield even smaller coverage than ubiquitination-proteomics, we conclude that our phosphoproteomic data here will only provide very limited information on PTM cross-talks such as such acetylation and ubiquitination. We therefore added a note in the Discussion, suggesting that “Other PTMs and PTM cross-talking events on top of phosphorylation will be also interesting to measure following optogenetic stimulation by combining PTM enrichment and DIA-MS in future studies.” (Page 16 Line 494).

Figure R2

2. In Figure 1F, it is not quite clear about the phospho-enrichment efficiency, please elaborate, and provide additional figures/tables as necessary.

Author reply: This better description is “phosphopeptide enrichment efficiency”. The text is now replaced (Page 6 Line 170).

3. It is not quite clear how many biological/technical replicates per group were acquired in the experiment, some of the error bars (e.g., Pe025 of AKT_T308 in figure 1G) are not trivial. The inter-sample CV% of each group should be provided in the supplemental material.

Author reply: Two biological replicates were used per condition, which was already mentioned at (now Page 5 Line 160, for this DIA-MS based investigation. We also add “n=2 biological replicates” into the legend of Figure 1G and Methods (Page 25 Line 856). Because of the biological duplicates, we have calculated the Pearson correlation coefficients of each group (instead of providing an inter-sample CV%), please see Figure S4A. While the .sne file is provided at PRIDE, the P-site specific quantification results can be inquired by the “Download Data Table” function in <https://yslproteomics.shinyapps.io/AKTPhos/>.

4. Can authors discuss more potential applications of optogenetics-proteomics approach in biomedical research.

Author reply: We thank the reviewer for this suggestion. We have provided more discussions for this request (Page 17 Line 503, pasted below).

“The optogenetics approach has shown promising therapeutic potential in treating neurological diseases and cancers^{13,14} by manipulating the activity of specific cells or proteins in disease models in a highly controlled manner, while proteomics and phosphoproteomics can be used to refine the optogenetic parameters, understand the molecular response to these manipulations, and identify potential drug targets and therapeutic opportunities to improve their safety and efficacy”

Reviewer #2 (Remarks to the Author):

The authors described a novel aspect of Akt signaling with combined use of an optogenetic system to activate Akt and quantitative phosphoproteomics techniques. Overall, the experimental design to decipher the temporal aspect of Akt signaling is sound and the obtained result provides potential candidates for novel Akt-mediated phosphosites and proteins. In addition, a Web application of Optop-DIA results makes public access to the authors' findings and contributes to the phosphosites research in the field of Akt signaling research. However, there are no data with other approaches to support their findings by the phosphoproteome analysis, which is critically important to validate the results. In addition, some of the results displayed in the figures are not easy to be interpreted and need to be clarified more. This reviewer felt the authors need to address the following concerns;

Author reply: We thank the reviewer for the constructive comments. As shown in the point-to-point reply below, we believe that we have greatly improved both bioinformatic and experimental validation for the newly discovered Akt substrate or associated P-sites in the revised manuscript, following this reviewer's suggestions (Major point 3). Moreover, we have strengthened and analysis and presentation of phosphoproteomic results.

Major concerns

1. Representation of data in Figures 2, 3, 4, and 5 is not easy to be interpreted and is not fully explained in the legend. Specifically, please consider the revision of the following figures.

Figure 2D: Can the authors provide scales for color coding of the heatmap? In addition, if there are no related GO terms for hC3, I think the authors can leave the section blank instead of listing detected phosphosites because they are not a GO term therefore somewhat misleading.

Figure 3B, Figure 4: The color coding for each line is unclear. What is the difference between blue and red? In addition, can the authors provide representative behavior for each cluster such as the average profile?

Figure 5B: Labels on the horizontal axis lacks information on the condition for what the data correspond to (is it time?). In addition, is it difficult to combine the result of two replicates for each condition? Also, please consider adjusting the scale for the vertical axis to a constant value for all the clusters.

Figure 5C: Please adjust the scale for each cluster. For instance, mC1 corresponds to phosphosites that decreased in phosphorylation in Ang stimulation and mC2 corresponds to phosphosites that increased in phosphorylation by Ang stimulation, but visually cannot be discriminated. In addition, what does the dotted line represents in the figure?

Author reply: Thanks for these concrete suggestions regarding the data presentation in Figures. We have now implemented all the suggested revisions.

- **Figure 2D:** We have implemented the requested change to add color bar and leave GO terms blank for hC3 and modified the legend accordingly.
- **Figure 3B/Figure 4:** We have added the description of color coding into the legend. Line colors were generated by the "Mfuzz" package of R, which denote the membership values after FCM clustering i.e., a similarity score of vectors to each cluster. The yellow-green-purple-red color palette corresponds to the low to high membership value. We now also clarified this in the Methods (**Page 30 Line 987**).

Because the FCM analysis was introduced as a soft clustering analysis with membership score reported that is designed to be better than the average profiles in representing clusters¹⁵, the Mfuzz package does not plot the averaged profiles. However, based on the reviewer's suggestion here, we have now included a new Supplementary **Figure S6** to show the averaged profiles for both Figure 3B and Figure 4. From Figure S6, one can see very similar patterns to Figure 3B and Figure 4.

- **Figure 5B/5C:** Following the reviewer's comments here, we have firstly improved the previous Figure 5B/5C but decided to move them into Supplementary **Figure S9** in the revision—These figures were generated by the "maSigPro" R package and, as the reviewer pointed out, are not friendly in seeing axis and trends. We replaced them with the **new Figure 5B** which is much simpler and more straightforward.

- **New Figure 5B:** We now use nested boxplot to visualize the regulations of P-sites in each maSigPro identified cluster (mC1-mC4). The reviewer is correct *re.* "mC1 corresponds to phosphosites that decreased in phosphorylation in Ang stimulation and mC2 corresponds to phosphosites that increased in phosphorylation by Ang stimulation". We have adjusted the corresponding description in both Results and Discussion (**Page 11 Line 331**, and **Page 15 Line 445**). Thank you for pointing this out!

- **Previous Figure 5B (now Figure S9A):** Yes, the horizontal axis used to lack information on the condition for what the data corresponds to time. And this is now added to the figure. Scale is now set uniform in the new Figure 5B boxplots.

- **Previous Figure 5C (now Figure S9B):** Because of the new Figure 5B, herein we kept the default scale provided by maSigPro^{16,17} to visualize the relative difference between curves. We however highlighted the $\log_2(\text{FC})=0.0$ position in the y

axis using a different color. In addition, we add the description of solid and dotted line into the figure legend: “Dots show actual fold-change values. Solid lines denote the average value of fold-change at each time point for each experimental group. Fitted curves after computed regression analysis using maSigPro¹⁶ are displayed as dotted lines.”

2. For the FCM analysis in Figure 3 and 4, I could not follow why the authors initially filtered phosphosites that were phosphorylated by the Pe025 condition. What would be the difference if filtered by Su025 and/or without filtering? The authors should explain what this filtration process is intended to be in the text. In addition, although I agree that phosphorylation for phosphosites in fC2 delays that in fC1, I think it is too exaggerated to call it a “waves of a signaling cascade” as there is no evidence of causative relationships in responsible kinases for fC1 and fC2 (in other words, the findings by FCM analysis only show that phosphorylation at phosphosites in fC1 and fC2 are timely distinct events). If the authors claim it is a cascade event, they need to clarify the causative relationships or activation time difference of at least one or two potential responsible kinases for those clusters.

Author reply: We took this point seriously, which, in general, as we understand, is about how to strengthen our phosphoproteomic analysis for elaborating causative relationships. In short, we now have utilized the state-of-the-art bioinformatic tools tailored for phosphoproteomics data analysis from both the Yang and Saez-Rodriguez labs (instead of clustering analysis only, see below) to identify potential responsible kinases and signaling network.

Firstly, with respect to the Pe025 condition based selection. We have already demonstrated the analysis of biological variance in the entire (i.e., no filtered) dataset in Figure 2. Because the 5-min length of darkness only reduces the p-Akt by >50% (rather than eliminating it, Figure 1D), we regard the periodic condition as the experiment reducing then stimulating p-Akt levels (at 10, 20, 30 mins, Figure 1F). The sustained condition might include more long-term P sites and feedback signaling events. We have now improved the explanation (**Page 8 Line 231-234**). Needless to say, this filtering step is just our approach to focus on a highly relevant P-site list attributable to Akt. In the revision, we now also consider all of the experimental conditions (intensity, pattern, and duration) across the experiment to predict the kinase-substrate relationships from this list by PhosR¹⁸ (see below).

Secondly, following the review’s comment, we removed the claim that we discovered “waves of a signaling cascade” throughout the text and revised the wording (e.g., **Page 8 Line 231, Page 9 Line 275**). The result section subtitle has been changed to “**Regulated phosphorylation events with variable decay following short-term AKT activation**”.

Thirdly, even if we removed the claim of “waves of a signaling cascade”, as the reviewer pointed out, we still indicate the sequential signaling events following short-term p-Akt activation. Therefore, in the revision, we have significantly improved the phosphoproteomics data analysis by combining the signaling reconstruction tool PHONEMeS¹⁹ from the Saez-Rodriguez lab and the kinase-substrate prediction tool PhosR¹⁸ from the Yang lab to explore the responsible kinases and signaling network following Akt activation. Please see the new Results at **Page 9 Line 263-270**, the new **Figure 3F** and the Methods (**Page 29 Line 952-977**).

Briefly,

- We set Akt1 to the first known kinase upregulated as the input for PHONEMeS¹⁹.
- We extract the subnetwork containing nodes 3 steps downstream of Akt1. By applying integer linear programming implementation for causal reasoning, PHONEMeS finds a path that connects a set of experiment-specific deregulated P-sites with deregulated kinases in a prior knowledge network²⁰.
- We use PhosR¹⁸ to predict kinase-substrate relationships for unique P-sites in clusters fC1-fC4 by combining the information about dynamic profiles of canonical substrates of kinases in all conditions across the experiment.
- We map the kinases predicted by PhosR and the subnetwork extracted by PHONEMeS, which is presented as the new Figure 3F.
- We found that the kinases and their relative positions in the network in Figure 3F largely agreed with Figure 3C and 3E and our refined “claim” that short-term Akt activation organizes temporally successive and functionally divergent processes that are separable from the initial Akt stimulus.

In summary, we feel that our strengthened phosphoproteomic analysis took the benefits of the unique combination of PHONEMeS and PhosR, two state-of-the-art, P-site based bioinformatic tools and revealed activation time difference of putative kinases.

3. One of the key findings in this study is the identification of novel Akt-associated phosphosites and proteins. If the authors could verify some of the newly discovered proteins with other approaches (i.e. biochemical), that would be strong supporting data for what the authors claim.

Author reply: Thank you for this suggestion. We believe that we have greatly improved both bioinformatic and experimental validation for the newly discovered Akt substrate or associated P-sites in the revised manuscript.

To facilitate presentation, we define our core phosphoproteomic data (i.e., those 112 P-sites showing $R > 0.85$ quantitative correlation to Akt p-T308) as the ***OptoCore*** list. These 112 P-sites are derived from 77 proteins (See Columns A-C and AU of **Table S1**).

1. ***Experimental approach.*** We decided to choose an elegant biochemical approach published by Knight et al ²¹ which is essentially an *in vitro* kinase assay in cell lysate using FSBA and quantitative mass spectrometry (**Figure S11** and Methods, **Page 27 Line 904-916**). This substrate-finding approach overcomes the obstacle of endogenous protein kinase activity by FSBA (5'-4-fluorosulphonylbenzoyl adenosine), an ATP analogue that inhibits protein kinases by occupying the ATP binding site and covalently attaching to an invariant lysine²². This approach has several particular advantages over alternative approaches – e.g., it can provide direct substrate identifications in a single experiment ²¹. Note, after FSBA treatment, the abundance of state-steady phosphoproteome remained the same, but all the kinases are deactivated by FSBA. Thus, the addition of Akt will not trigger a global downstream phosphoproteome change but only the phosphorylation of Akt direct substrates. Herein, we combined the assay with the plexDIA method using mTRAQ labeling ²³ before phosphopeptide enrichment (**Figure S11**).

We used both western blots as well as quantitative phosphoproteomic analysis by DIA-MS to interrogate the result of the kinase assay (new **Figure 6**). To validate our OptoCore list in more than one EC cell line, we included both the normal (**i.e., non-transduced**) EA.hy 926 cells and HUVEC cells in this experiment.

- ***Western blotting:*** Please see the new **Figure 6G-H**. We first confirmed that the classic Akt substrate P-sites (eNOS_S1177 and GSK3 β _S9) were upregulated after adding Akt and ATP into the FSBA-deactivated cell lysate compared to the controls where only PBS were added. Next, we compared our OptoCore list with the available commercial antibodies. We identified two antibodies available for two novel P-sites, NEDD4L_S448 and NRG1_S330, that were not reported as AKT substrates in PhosphositeDB. Western blotting confirmed the upregulated phosphorylation for both sites, validating that OptoCore enriched novel Akt substrate P sites.

- ***Plex-DIA:*** Please see the new **Figure 6E-F**. To our knowledge, our new validation MS dataset presents the first application of PlexDIA based on mTRAQ labeling (PlexDIA is a DIA-MS quantification combined with the chemical mTRAQ labeling for different samples, See ²³) in phosphoproteomics. By combining the results from DIA-NN and Spectronaut, we were able to quantify the 6565 and 9468 ratio values for the phosphopeptides in the FSBA based in kinase assay in the EA.hy 926 cells and HUVEC cells respectively. Though the coverage of P sites was much less than our Opto-DIA dataset, encouragingly, the ratio values of the OptoCore list (n=50 and 61 in EA.hy 926 cells and HUVEC cells) are remarkably higher with Akt treatment compared to the control cell lysates ($P < 2.2e-16$, for both EA.hy 926 cells and HUVEC cells, **Figure 6E**). Moreover, we were able to quantify 19 unique P sites in OptoCore list from both cell lines, all of which in EA.hy 926 cells and 18 out of 19 in HUVEC cells were upregulated after Akt treatment (**Figure 6F**) (2.98-fold and 4.02-fold upregulated on average).

In summary, following the reviewer's comment, we have now provided strong experimental evidence, using both a biochemical approach and MS, verifying some novel P sites as direct Akt substrates.

2. ***Bioinformatic approach.*** This was done by a comprehensive comparison between our OptoCore list (n=112 P-sites) to a series of state-of-the-art datasets to interrogate the specificity and quality of our dataset. These four lists are:
 - a. The 342 P-sites listed as Akt substrate in Phosphosite database (We identified 145 P-sites. Hereafter, ***PsPdb***).
 - b. The 1033 P-sites we identified carrying the Akt motif R-x-R-x-x-S*/T* (Hereafter, ***Akt motif***)
 - c. The 153 we identified that were effectively downregulated P-sites upon treatment with all five Akt inhibitors, according to a recent study from Kuster lab ¹¹ (Hereafter, ***5 Inhibitors***)
 - d. The 257 P-sites which has an Akt "kinome score" with the percentile ≥ 0.99 , according to a recent atlas study published at Nature using synthetic peptide libraries and computational prediction ¹² (Hereafter, ***Kinome***)

We summarized the comparison result in **Figure 6A-D**. Please see the Results text at **Page 12 Line 363-377**. Basically, our OptoCore list, despite its smaller proteome coverage (as a single study), showed a high consistency to all of the four lists (**Figure 6A**). Also OptoCore reached a high *selectivity* comparable to PsPdb, both of which are better than the other three lists (**Figure 6C**). This means, although a large fraction of P-sites in our OptoCore list was not reported by PsPdb (**Figure 6B**), they are high-quality putative Akt substrate P-sites and exactly present the value of OptoCore.

Minor concerns

1. Although the authors selected light illumination conditions and duration to avoid cell toxicity, the effect of light on the phosphoproteome is elusive. Can the authors provide information on the influence of light stimulation on phosphoproteome?

Author reply: We now performed a new experiment as requested. We stimulated the parental EA.hy 926 and HUVEC endothelial cells (i.e., non-transduced cells) with the blue light of our highest intensity and duration condition (i.e., 0.25 mW/cm² for 30 mins) and performed phosphoproteomics DIA-MS measurements. With three biological replicates, we could not detect any global variance higher than the variance between replicates, i.e., cells under blue light stimulation did not cluster together in the heatmap (new **Figure S3A**), which is in stark contrast to Figure 2A. As expected, no abundance difference was observed for individual classic Akt substrates (such as those P-sites in Figure 1G). This experiment thus confirmed that the blue light we used was indeed extremely low.

This result is not surprising. Basically our highest and most intensive condition of blue light (i.e., 0.25 mW/cm² for 30 mins) was similar to the “control” condition in those studies addressing the phototoxicity of blue light in the mammalian cell line⁵. Please also refer to the Reply to Reviewer 1 (Point 1).

2. The interesting finding in Figure 3 is that phosphosites for fC3 are enriched in the MAPK phosphorylation motif. Because the authors state that one of the kinases of the MAPK, ERK, is not activated by optogenetic Akt activation, what could be the responsible kinase for fC3? Can the authors provide data to support their findings, such as presumable activation of ERK, p38, and Jun (*Author note: we guess the reviewer meant JNK here*) from their phosphoproteomics measurements?

Author reply: We found that optogenetic stimulation of AKT induced only a low level of ERK phosphorylation (e.g., MAPK3_Y204, a classic ERK regulatory P-site), especially when compared to robust ERK phosphorylation observed with Ang1 treatment – Please see **Figure 5D** (MS result) and **Figure S8A** (western blotting result). However, many downstream P-sites are overlapping between Akt and MAPK pathways, which was the reason we performed the relative kinase enrichment analysis of **Figure 5C**. Our improved phosphoproteomic analysis using PHONEMeS and PhosR indicate the potential activation of MAP3K8 (**Figure 3F**), which activates both the MAP kinase and JNK kinase pathways²⁴. In addition, both **Figure 3F** and **Figure 5C** suggested an activation of PRKACA (or PKA), which can promote p38 MAPK activation²⁵. Furthermore, some established substrates of both ERK and p38 were found among the fC3/4 sites, including the well-characterized p38 substrates MEF2A/C. By contrast and in keeping with established inhibitory crosstalk between AKT and JNK signaling²⁶, we did not detect phosphorylation of unique JNK substrates. However, many key “index” MAPK target sites present in the full dataset were not induced by optogenetic AKT activation, and it is not clear which kinase is driving the enriched MAPK family motifs. Furthermore, we did not obtain direct evidence to determine whether ERK or p38 are individually or coordinately activated. Due to the complexity of overlapping P-sites based on motif features for different kinases and the potential cell specificity issue, we do not believe we can confidently infer the preferable activation between ERK, p38, and JNK from the current dataset which is focusing on Akt signaling.

3. In Figure 3D, can the authors also provide “%with Akt motif” for entire detected P-sites to represent enrichment of Akt motif in fC1?

Author reply: Yes. This result was actually already included in Table S1, which presents the entire dataset. Among the total of 34,740 P-sites measured, 1033 of them (i.e., 2.97%) carries the Akt motif, which is similar to the percentages of fC2-fC4. We have added this information in the Result text describing Figure 3D. Please see **Page 9 Line 257**.

4. Wherever possible, please consider indicating the specific phosphosites in the text and figures because identification of phosphosites is one of the key findings of this research. For instance, in Figures 1D and E, it would be beneficial to present specific phosphosites of eNOS and GSK3 β .

Author reply: This is an excellent suggestion. We have now added all the phosphosite information into all western blotting Figures (e.g., Figure 1, Figure S1, Figure S2, new Figure S8) and relevant text.

5. For the enrichment analysis, the authors use motif enrichment for some cases and annotated kinase enrichment analysis for other cases. What are the criteria for selecting the type of enrichment analysis? Could the authors perform both types of enrichment analysis?

Author reply: For **Figure 3**, we are particularly interested in extracting potential motifs in fC1-fC4 and related that to sequential signaling events in the cell. However, as noted above, the motif analysis has many limitations, such as the overlapping P sites between kinases. For example, we identified 1033 P sites carrying Akt substrate motif, but there are only 50 of them (i.e., only 4.84%) are supported by our OptoCore list. Please note that we have provided the entire dataset of our study in Table S1 and our Web application, which can support all the possible analyses including motif enrichment analysis, P-site based kinase annotation,

P-protein based annotation, PHONEMeS, or PhosR or other analyses. We unfortunately could not perform and discuss all the possible results in this rich resource present in the study.

6. In Figure 5D, what does the asterisk for “GSK3B*” mean?

Author reply: Sorry that we missed this information in the legend, which is now added in the revision. Basically, in the mC3&4 circle, the asterisk following GSK3B denotes that only GSK3B is enriched from mC4 whereas other kinases are enriched from mC3.

7. In L312, the authors describe mC3 & mC4 as “cross-talking nodes”, however, because they are the phosphosites both regulated by optoAkt and Ang, I think they are “Akt-regulated nodes”. I’m curious why the authors stated they are cross-talking.

Author reply: The concept of “pathway crosstalk” was widely used in many biological settings to describe instances in which two functional pathways interact with each other since certain biomolecules may have more than one role and may be involved in more than one biological function²⁷. In the previous version of our manuscript, we tended to follow this broad concept and used the term in a non-specific manner without a definition (as Reviewer 3 also pointed out) to describe the crosstalk between ERK (i.e., not Ang1) and Akt pathways²⁸. Indeed, many different researchers have different preference and interpretation of using the concept of “pathway crosstalk”²⁹, we therefore decided to eliminate the usage of the word “cross-talk” in the revision. Instead, we now refer to mC1 and mC2 as “either relatively downregulated or upregulated by Ang1” whereas mC3 and mC4 as “common P-site usage in AKT and growth factor signaling” (**Line 331-334, Page 11**). The title of this Result section reads now “**Identifying P-sites commonly and preferably regulated by growth factor and AKT signaling**”.

8. In Figure6, the rules for the search of Akt substrate phosphosites are interesting. Can the authors provide information on the number of phosphosites that matched each rule, and also whether there was any overlap of phosphosites among phosphosites that matches each rule?

Author reply: As Reviewer 3 pointed out that the known Akt motif is just “an essential feature of bona fide Akt substrates”, we no longer use the loose motif as a filter anymore in Figure 6. The new **Figure 7** (previously Figure 6) just serves as a nice overview of OptoCore P sites which also carries an essential Akt motif of R-X-R-X-X-S/T for some readers. We however extensively compared our OptoCore sites to known high-quality datasets (including the Akt motif) in the new **Figure 6** (see above). For the overlap of phosphosites, please refer to **Figure S10A**.

9. Several typos are found in the text.

L157: “p-Akt T308” should be “Akt p-T308”.

L304: “Su020” should be “Su025”.

Author reply: Thank you for catching these typos. They have now been fixed.

Reviewer #3 (Remarks to the Author):

Summary

The authors of this manuscript present a new workflow for the analysis of Akt signaling in endothelial cells (EC) in which they combine a previously reported optogenetic system (ref #30) for controlling Akt translocation to the plasma membrane with data-independent acquisition mass spectrometry (DIA-MS) (ref #18). As previously reported, the authors demonstrate that the optogenetic system works robustly to permit the light-dependent, reversible translocation of Akt to the plasma membrane. The authors achieve impressive technical results in their DIA-MS pipeline regarding coverage of the phospho-proteome, and have generated a wealth of data regarding the changes in phospho-site abundance following light-stimulated Akt translocation. Using hierarchical clustering analysis and principal component analysis, the authors classify phospho-sites that are upregulated into different groups according to the temporal signatures of the phospho-site abundance. In general, the work is technically well performed, but the insights and conclusions are somewhat limited. As a data resource, the work may (if revised adequately) represent a useful cross-validation tool for published and future studies, while the pipeline has the potential to compare Akt signaling in different cell lines under different stimulation regimes.

Author reply: As a general reply, we would like to acknowledge the reviewer for pointing out the technical advance of our phosphoproteomic dataset and the opportunity for us to improve the description, presentation, and the discussion about our optogenetic experiment system. Please see the point-by-point reply below in which we also made an important clarification that the motif filtering strategy was solely used for the previous Figure 6 (now revised) to visualize some P sites. The core contribution of our data is not relevant to the current knowledge of Akt motif. Also, we have improved both bioinformatic and experimental validation for our dataset in the revised manuscript, exactly strengthening the reviewer's comment here about "a useful cross-validation tool".

Major comments

Introduction / experimental design

The authors do not cite recent and accumulating evidence that Akt is dependent on PIP3 or PI(3,4)P2 for its activity, not just on its phosphorylation by PDK1 and mTORC2. This is important, since the authors cite their own data on page 4 that endogenous PI3K activation is required for light-stimulated Akt activation.

Author reply: We apologize for omitting reference to this recent work, primarily from the Leonard laboratory, that is directly relevant to our manuscript (discussed further below). However, we would also like to point out that other work from Phil Cole's laboratory suggests that C-tail phosphorylation rather than lipid engagement is sufficient to reverse autoinhibition by the PH domain. In our minds the issue of how and where AKT is found in its active state has yet to be completely resolved, and we feel it important to cite both lines of evidence. We have added additional text to the introduction with appropriate citations (**Page 2 Line 58**):

"For example, growth factor and insulin signaling lead to the activation of class I phosphoinositide 3-kinase (PI3K), which generates the lipid second messenger phosphatidylinositol-3,4,5-trisphosphate [PI(3,4,5)P₃, or PIP3]. AKT is recruited via its PH domain to PIP3 and phosphatidylinositol-3,4-bisphosphate [PI(3,4)P₂] in the plasma membrane (PM)³⁰⁻³³. The recruitment of AKT to PM promotes its phosphorylation by phosphoinositide-dependent kinase-1 (PDK1) and the mechanistic target of rapamycin complex-2 (mTORC2) at residues Thr308 and Ser473^{34,35}, which are essential for AKT catalytic activity. In addition, recent evidence suggests that binding to phosphoinositides relieves autoinhibition of the catalytic domain by the PH domain, suggesting that AKT is only active and capable of phosphorylating substrates when membrane-bound^{32,33,36}. Alternatively, it has been proposed that mTORC2 phosphorylation alone can relieve these autoinhibitory constraints^{37,38}"

The authors do note in their discussion that their reporter is limited to Akt signaling at the PM, whereas numerous reports have indicated Akt signaling on endomembrane compartments. However, some aspects of their reporting would benefit from additional clarification. Specifically:

(a) Their reporter is not light-activated, as they claim/imply. They use light to control the subcellular localization of Akt, not its activation. As they show themselves, and which is supported by numerous reports, Akt activation depends on PI3K and the lipid second messenger PIP3. As such, their reporter reports on the endogenous pathways that activate PI3K, albeit amplified by increasing the residence time of Akt at the PM. Essentially, the authors have designed a reporter that can interrogate the activity of Akt1 at the PM, since they anchor it there with a prenylated lipid anchor. This means that the signaling pathways triggered downstream of Akt are originating from PIP3-activated, PM-resident Akt, and not Akt that is active on any PIP3- or PI(3,4)P2-

containing endomembrane in the cell. The authors should clarify this in their discussion (pg 14 ln 412-416 should be expanded with additional citations). Given the context in which Akt is activated (i.e. by endogenous PIP3), it is questionable what useful information is provided by the differing light stimulation regimes, which induce a non-physiological, reversible translocation of Akt from the cytosol to the PM. What the authors are, in fact reporting on, is a convolution of endogenous PI3K signaling with an artificial, light-dependent recruitment that serves to transiently increase Akt residence times at the PM, during which Akt can be activated more readily.

(b) The recruitment of Akt to the PM using prenylated GFP is an artificial, perturbative mechanism of recruitment and the authors should acknowledge that this has the potential to distort the signaling pathways activated by this lipid-anchored Akt.

Author reply: We appreciate the reviewer’s comments on the value and the limitations of our optogenetic system. Indeed, as the reviewer would also acknowledge, any optogenetic system is “artificial” and discussion around this point would be very useful for our paper. The obvious benefit of using the optogenetic system is that it allows us to isolate the effect of AKT activation from other pathways that would be activated with a “native” stimulus, and allows for acute rapid activation of the kinase (consequent to membrane recruitment). As the reviewer is aware, there is a long history in the field of using artificially myristoylated AKT constructs³⁹ targeted to the plasma membrane as a “constitutively active” form of the kinase, and furthermore that this construct in at least some contexts supports AKT-dependent cell survival and promotes phosphorylation of key substrates. It is worth noting that many well-characterized substrates of AKT are not associated with the PM or endomembranes harboring PIP3 or PI(3,4)P2. So even though AKT can be activated at endomembranes in response to growth factors, it is reasonable to ask to what extent this is important for accessing substrates. Because activation of our light-controlled AKT occurs at the PM (and might, as the reviewer suggests, require continued association with the PM for activity), and because we have also conducted phosphoproteomic analysis in the same cell line with a “natural” stimulus (angiopoietin), we can actually try to address this question with respect to reported AKT substrates. The figure below (new **Figure S10E**) includes all reported AKT substrates catalogued in the PhosphoSitePlus database that were detected under all conditions in our dataset:

Figure S10E

Sites are divided into those correlating ($R > 0.5$) with AKT1-Thr308 phosphorylation across all conditions and are thus associated with AKT activity, and those that do not correlate and thus not phosphorylated by optogenetically controlled AKT. Each group is categorized into those that are upregulated by angiopoietin (Ang) treatment and those that conform to an AKT motif (either R-x-R-x-x-S/T or scoring at $>94.5\%$ based on peptide library analysis in Johnson et al.⁴⁰). As is evident, most substrates phosphorylated in response to Ang also correlate with AKT1-Thr308 phosphorylation well across all conditions (directly or indirectly) by opto-AKT. Notably, this set includes a number of well-characterized substrates that do not localize to the cell periphery – FOXO1/3, BAD, TBC1D4 (aka AS160), and AKT1S1 (PRAS40) among them. Most reported substrates that did not correlate with AKT1-pThr308 likewise did not become phosphorylated in response to Ang, and these sites were either erroneously reported as AKT substrates or are possibly not AKT substrates in this particular cell type. Collectively, this analysis suggests that the large majority of direct AKT substrates are indeed phosphorylated by opto-AKT. However, we note some exceptions, including the well-characterized substrate MDM2. Unexpectedly, two “Ang only” substrates localize to the plasma membrane (PLC- γ , SLC7A11/xCT), but none localize exclusively to endomembranes. We conclude that a minority of AKT substrates do indeed escape phosphorylation by optogenetically-controlled AKT1 with no clear correlation with protein localization. We have

now included additional discussion to acknowledge the reviewer's criticism in light of these observations and the historical use of myr-AKT.

For the revised discussion, please see **Page 16, Line 475-485**, which reads as following.

“One technical caveat for any optogenetic control system is that it cannot emulate all the steps during natural signaling. In particular, the optogenetic used in the present study essentially utilized the CRY2-CIBN dimerizer system to recruit AKT to PM to facilitate AKT phosphorylation by endogenous pathways. This means that, in the present system the AKT signaling originates from PIP3-activated, PM-resident AKT, but not AKT that is active on other PIP3- or PI(3,4)P2-containing endomembrane within the cell, whereas previous reports indicated AKT signaling also on cellular endomembrane compartments⁴¹⁻⁴³. We note however, that key downstream substrates of AKT not associated with the cell periphery were detected in our dataset, including the mitochondrial pro-apoptotic protein BAD and FOXO transcription factors. Indeed, the large overlapping P-sites between our OptoCore list and other high-quality datasets and our verification experiments independent of Opto-AKT together demonstrate the quality and specificity of the OptoDIA result.”

For the new validation experiments that are totally independent on Opto-Akt system (western blotting and new MS measurement) and bioinformatic comparison between OptoDIA result and other lists, please kindly refer to our Reply to the 3rd point of Reviewer 2, as well as our new **Figure 6** and **Figure S10, S11** and Result text (**Page 12-13**).

Results

The authors use motif enrichment analysis to identify Akt substrates based on a ‘strict’ Akt consensus motif and a ‘loose’ Akt consensus motif. The ‘strict’ motif, however, is simply an AGC kinase consensus motif, of which there are 63 in the human proteome. It is also erroneously cited as being R-X-R-X-X-S/T when in fact the minimal Akt recognition motif was reported to contain a bulky hydrophobic in the P+1 position. Using this motif as a filter will undoubtedly lead to many false positives. The ‘loose’ motif, however, is even more problematic, since Akt is known to depend on Arg in the P-5 position of its substrates (it forms a specific salt bridge with Glu279 in the kinase domain, Yang et al., NSMB 2002) and its reporting as an Akt consensus motif is based on questionable cell biological data. The authors of the review article cited by the authors (Manning and Toker, Cell 2007) considered (as I do), on balance of all the available data, an R-X-R-X-X-S/T motif to be an essential feature of bona fide Akt substrates.

Author reply: We acknowledge the reviewer's criticism, in particular the extremely generic nature of what we had referred to as the “loose” motif. However, we do not entirely agree with the reviewer's stance that all true AKT substrates conform to a strict R-x-R-x-x-S/T- ϕ sequence. The Manning/Toker 2017 review cited does articulate that definition, but at the same time acknowledges known substrates that lack the -5 Arg residue. Indeed, shortly following publication of that review, the Toker group reported the discovery of SLC7A11/xCT Ser26 (sequence: NGRLPSL) as an AKT substrate⁴⁴. Furthermore, the set of AKT substrates shown in Fig 2 of the 2017 Cell review include many lacking a +1 hydrophobic residue. We do concur that the R-x-R-x-x-S/T signature is shared with a subset of other AGC kinases, in particular S6K1 which is activated downstream of AKT. Accordingly, it is inappropriate to infer direct substrates by simply matching this consensus sequence to sites in our dataset.

We would however like to make important clarifications about the minimally way in which we used “Akt motifs”, since the “filtration strategies” in the previous Figure 6 seemed to cause a misunderstanding. Actually, we used our own data generated in this present study to identify Akt responsive P sites –including, but not restricted to, direct ATK substrates - To elaborate, there are two places where we used AKT motif information – our Figure 3C and old Figure 6 (the previous filtering step). In Figure 3C, our motif enrichment analysis essentially asked what sequences are enriched in each cluster among fC1-fC4, as compared to random scenarios. This means, our motif-enrichment analysis is hypothesis-free, i.e., we did not apply any prior knowledge and the analysis itself reported what we had referred to as strict and loose Akt motifs (or, R-X-R-X-X-S/T and R-X-X-S/T) as the top 2 mostly enriched motifs in fC1. While the reviewer is correct that we cannot claim that each site driving enrichment of these motifs are direct AKT substrates, the unbiased recovery of signatures associated with AKT is strongly suggestive that fC1 is enriched for AKT substrates. While we agree with the reviewer that both the strict and loose Akt motifs are not perfect, we did not rely on these motifs as being diagnostic for AKT substrates for Figure 3C.

In the previous Figure 6, we used these motif definitions as a concise summary to visualize those important P-sites that could be “Akt closely-associated P-sites and substrates” (see the previous Results subtitle). In that figure, we applied three rules in which we request the phosphosites to either have a quantitative correlation to Akt ($R > 0.85$ across all 31 conditions, which is termed **OptoCore** list in the revision) or fall into a relevant cluster (fC1, afC5 or afC11). The filter of strong quantitative correlation to Akt using our experiments can be crucial – for example, there were **1033 out of 34,740** human P-sites we detected carrying the strict motif among the total phosphoproteome we measured, whereas only **~50 P-sites** have an $R > 0.85$ based on our data ($50/1033=4.84\%$).

Because of the confusion due to our description of “rules”, we have now abandoned this wording of “filtering/ rules” and revised our text and also the new **Figure 7**, which now simply visualize P-sites in OptoCore list carrying the R-X-R-X-X-S/T motif. Notably, we no longer mention the ‘loose’ motif here. Again, this figure is only intended to highlight some interesting P-sites to (some) readers. The full dataset, including all OptoCore list and the full list of 34,740 P-sites are provided in Table S1 and Web application, with sites annotated based on whether they conform to the AKT consensus sequence.

To summarize, we agree with the reviewer that the current Akt consensus motif is imperfect; exactly because of its imperfectness, our study and analysis are valuable.

This being said, the reviewer’s comment here prompted us to examine the amino acid distribution frequency in reported Akt substrate P sites in PhosphoSitePlus database and list of sites strongly correlated ($R > 0.85$) with AKT1-Thr308 phosphorylation, which we term our **OptoCore** list:

Figure S10D

We note a high similarity of amino acids surrounding the P sites between PSP and OptoCore lists (mentioned now **at Page 12 Line 377-378** and shown in **Fig S10D**, pasted above). Indeed, OptoCore is actually more strongly enriched for sites with hydrophobic residues at the +1 position than are reported sites.

Pg 5, ln 139-140. The authors report the phosphoproteomic analysis of cells in duplicates. It is not clear from this or the methods section whether these ‘duplicate’ samples were true biological replicates or technical replicates of the same cells. This is important since, if they are technical replicates, the authors are reporting only one biological experiment on one cell line. Moreover, since the reporter actually reports on endogenous Akt activation by PIP3 (see pg 4, ln 108-111), considerable heterogeneity is likely to exist between individual cell populations grown under the same conditions, but at different times and in (potentially) different batches of growth medium. Whilst it is a significant amount of work, at least two biological replicates should be analyzed to get a sense of the reproducibility of the signaling networks activated downstream of Akt in endothelial cells.

Author reply: We agree and would like confirm that these are biological replicates (N=2) starting from initial cell culture (**Page 25 Line 856, Page 5 Line 160**). We hope our new validation experiments (western blotting and new MS measurement) and bioinformatic comparison further confirmed the reproducibility of OptopDIA dataset.

Minor comments

Pg. 7. P-sites classified in hC1 that persist for >30 min after switching off the light (and p-Akt with it) are not likely to be Akt substrates, but the consequences of signaling pathways upregulated by transient Akt activation. The authors imply that they could be Akt substrates in the nucleus, but this would necessitate them being phosphorylated at the PM, where the authors’ reporter is localized. The authors should clarify that these ‘substrates’ are unlikely to be direct Akt substrates.

Author reply: This is a good point. We added the clarification at **Page 7 Line 218-221**.

“Given that the active pool of AKT in the optogenetic system is likely restricted to the PM, the observed enrichment for AKT targets in the nucleus might indicate that nuclear targets in fact become phosphorylated in the cytoplasm as has been proposed by others⁴⁵”

The manuscript generally contains a lot of unnecessary, inflationary and emotional verbiage that is not scientific and should be avoided. Examples include, but are not restricted to:

Pg. 2, ln 64-65: “how cells decode the same signaling input from Akt and that result in distinctive downstream signaling defines a quintessential problem in modern molecular biology that remains largely unknown.”

Pg. 3, ln 77: “Conceivably, it would be straightforward and exciting to...”

Pg. 5, ln 142: “Using our high-performance phosphoproteomics-DIA...”

Pg. 11, ln 325 “Due to the favorable reproducibility...”

Pg. 9, ln 277: “...representing an uncharted complexity in Akt signaling.”

Pg. 8, ln 234: “...remarkably enriched in...”

Pg. 9, ln 267: “afC6 is fascinating as well, because...”

Pg. 7, ln 214: “...exhibits intriguing behaviors.”

The authors use the words ‘dynamics’ and ‘cross-talk’ repeatedly in a very unspecific manner which is not helpful to the reader.

The manuscript contains a large number of typos, spelling, punctuation and grammatical errors.

Author reply: We are sorry for the usage of emotional wordings and the terms. We now tried our best to eliminate them and to improve the textual and language with these issues.

1. Following the sequence the Reviewer mentioned, here are how the new text reads.

“How cells decode the *same* signaling input from AKT into *distinctive* downstream signaling remains largely unknown” (**Line 74**)

“One way to provide a more global view of signaling output from a specific pathway would be to combining optogenetic tools with mass spectrometry (MS)-based phosphoproteomics ...”(Line 93)

“Using phosphoproteomics-DIA...” (Line 163)

“Due to the reproducibility of DIA-MS” (Line 404)

The “uncharted” sentence is now removed (Line 425)

“Remarkably” is removed. (Line 286)

The “fascinating” sentence is removed. (Line 300)

“exhibits different behaviors” (Line 238)

We also tried to remove the emotional wordings in other places such as **Line 171** and many other places.

2. “Dynamics”: We have revised most “dynamics” to “temporal dynamics” or “regulation” to be more specific.

3. “Cross-talk”: The concept of “pathway crosstalk” was widely used in many biological settings to describe instances in which two functional pathways interact with each other since certain biomolecules may have more than one role and may be involved in more than one biological function²⁷. In the previous version of our manuscript, we tended to follow this broad concept and used the term in a non-specific manner without a definition (as this Reviewer pointed out) to describe the crosstalk between ERK (i.e., not Ang1) and Akt pathways²⁸. Indeed, many different researchers have different preference and interpretation of using the concept of “pathway crosstalk”²⁹, we therefore decided to eliminate the usage of the word “cross-talk” in the revision. Instead, we now refer to mC1 and mC2 as “either relatively downregulated or upregulated by Ang1” whereas mC3 and mC4 as “common P-site usage in AKT and growth factor signaling” (**Line 331-334, Page 11**). The title of this Result section reads now “Identifying P-sites commonly and preferably regulated by growth factor and AKT signaling”

4. In the revision, we tried our best to fix all typos, spelling, punctuation and grammatical errors.

Figure 3B and Figure 4. There is no legend provided to explain the color code used in these panels. Presumably, each line represents an individual phospho-site, but the meaning of the different colors is unclear.

Author reply: Thanks. We have added the description of color coding into the Legend. Basically, the colors of lines were generated by the “Mfuzz” package of R, which denote the membership values after FCM clustering i.e., a similarity score of vectors to each cluster. The yellow-green-purple-red color palette corresponds to the low to high membership value for individual P-site. We also clarified this in the Methods (**Page 30 Line 987**).

Supplementary Figure 1 legend. The legend to panel B is non-sensical and needs considerable revision.

Author reply: We have improved the legend of Figure S1B. See below.

“(B) The Akt1 R25C carried an amino acid mutation at the PH domain of Akt1 that blocks the Akt1 from binding with the plasma membrane and therefore cannot be activated by the upstream. The Akt1 K179M mutated the ATP-binding site 179 of Akt, which leads to kinase defective activity. The Western blot here demonstrated that there was decreased activities of exogenous Akt1 with the mutations of K179M and Akt1 R25C following light stimulation and 20% FBS, whereas the endogenous Akt1 can be always activated by 20% FBS despite variable Opto-Akt mutations”.

Supplementary Figure 5C. There are more rows to the table of data presented than there are GO terms to annotate them. The missing terms should be added and the terms aligned properly with the table rows.

Author reply: Thank you for catching this up. Indeed, this was due to our error during figure preparation and is now fixed.

References

1. Xu, Y., Nan, D., Fan, J., Bogan, J.S., and Toomre, D. (2016). Optogenetic activation reveals distinct roles of PIP3 and Akt in adipocyte insulin action. *J Cell Sci* 129, 2085-2095. 10.1242/jcs.174805.
2. Kennedy, M.J., Hughes, R.M., Peteya, L.A., Schwartz, J.W., Ehlers, M.D., and Tucker, C.L. (2010). Rapid blue-light-mediated induction of protein interactions in living cells. *Nature methods* 7, 973-975. 10.1038/nmeth.1524.
3. Yazawa, M., Sadaghiani, A.M., Hsueh, B., and Dolmetsch, R.E. (2009). Induction of protein-protein interactions in live cells using light. *Nature biotechnology* 27, 941-945. 10.1038/nbt.1569.
4. Emiliani, V., Entcheva, E., Hedrich, R., Hegemann, P., Konrad, K.R., Lüscher, C., Mahn, M., Pan, Z.-H., Sims, R.R., Vierock, J., and Yizhar, O. (2022). Optogenetics for light control of biological systems. *Nature Reviews Methods Primers* 2, 55. 10.1038/s43586-022-00136-4.
5. Alghamdi, R.A., Exposito-Rodriguez, M., Mullineaux, P.M., Brooke, G.N., and Laissue, P.P. (2021). Assessing Phototoxicity in a Mammalian Cell Line: How Low Levels of Blue Light Affect Motility in PC3 Cells. *Front Cell Dev Biol* 9, 738786. 10.3389/fcell.2021.738786.
6. Ong, Q., Guo, S., Duan, L., Zhang, K., Collier, E.A., and Cui, B. (2016). The Timing of Raf/ERK and AKT Activation in Protecting PC12 Cells against Oxidative Stress. *PloS one* 11, e0153487. 10.1371/journal.pone.0153487.
7. Liu, Y., Beyer, A., and Aebersold, R. (2016). On the Dependency of Cellular Protein Levels on mRNA Abundance. *Cell* 165, 535-550. 10.1016/j.cell.2016.03.014.
8. Liu, Y., Borel, C., Li, L., Muller, T., Williams, E.G., Germain, P.L., Buljan, M., Sajic, T., Boersema, P.J., Shao, W., et al. (2017). Systematic proteome and proteostasis profiling in human Trisomy 21 fibroblast cells. *Nature communications* 8, 1212. 10.1038/s41467-017-01422-6.
9. Huttlin, E.L., Bruckner, R.J., Navarrete-Perea, J., Cannon, J.R., Baltier, K., Gebreab, F., Gygi, M.P., Thornock, A., Zarraga, G., Tam, S., et al. (2021). Dual proteome-scale networks reveal cell-specific remodeling of the human interactome. *Cell* 184, 3022-3040 e3028. 10.1016/j.cell.2021.04.011.
10. Szklarczyk, D., Gable, A.L., Nastou, K.C., Lyon, D., Kirsch, R., Pyysalo, S., Doncheva, N.T., Legeay, M., Fang, T., Bork, P., et al. (2021). The STRING database in 2021: customizable protein-protein networks, and functional characterization of user-uploaded gene/measurement sets. *Nucleic Acids Res* 49, D605-D612. 10.1093/nar/gkaa1074.
11. Wiechmann, S., Ruprecht, B., Siekmann, T., Zheng, R., Frejno, M., Kunold, E., Bajaj, T., Zolg, D.P., Sieber, S.A., Gassen, N.C., and Kuster, B. (2021). Chemical Phosphoproteomics Sheds New Light on the Targets and Modes of Action of AKT Inhibitors. *ACS chemical biology* 16, 631-641. 10.1021/acscchembio.0c00872.
12. Johnson, J.L., Yaron, T.M., Huntsman, E.M., Kerelsky, A., Song, J., Regev, A., Lin, T.-Y., Liberatore, K., Cizin, D.M., Cohen, B.M., et al. (2022). A global atlas of substrate specificities for the human serine/threonine kinome. *bioRxiv*, 2022.2005.2022.492882. 10.1101/2022.05.22.492882.

13. Chen, W., Li, C., Liang, W., Li, Y., Zou, Z., Xie, Y., Liao, Y., Yu, L., Lin, Q., Huang, M., et al. (2022). The Roles of Optogenetics and Technology in Neurobiology: A Review. *Front Aging Neurosci* *14*, 867863. 10.3389/fnagi.2022.867863.
14. Malogolovkin, A., Egorov, A.D., Karabelsky, A., Ivanov, R.A., and Verkhusha, V.V. (2022). Optogenetic technologies in translational cancer research. *Biotechnol Adv* *60*, 108005. 10.1016/j.biotechadv.2022.108005.
15. Futschik, M.E., and Carlisle, B. (2005). Noise-robust soft clustering of gene expression time-course data. *J Bioinform Comput Biol* *3*, 965-988. 10.1142/s0219720005001375.
16. Conesa, A., Nueda, M.J., Ferrer, A., and Talon, M. (2006). maSigPro: a method to identify significantly differential expression profiles in time-course microarray experiments. *Bioinformatics* *22*, 1096-1102. 10.1093/bioinformatics/btl056.
17. Nueda, M.J., Tarazona, S., and Conesa, A. (2014). Next maSigPro: updating maSigPro bioconductor package for RNA-seq time series. *Bioinformatics* *30*, 2598-2602. 10.1093/bioinformatics/btu333.
18. Kim, H.J., Kim, T., Hoffman, N.J., Xiao, D., James, D.E., Humphrey, S.J., and Yang, P. (2021). PhosR enables processing and functional analysis of phosphoproteomic data. *Cell reports* *34*, 108771. 10.1016/j.celrep.2021.108771.
19. Gjerga, E., Dugourd, A., Tobalina, L., Sousa, A., and Saez-Rodriguez, J. (2021). PHONEMeS: Efficient Modeling of Signaling Networks Derived from Large-Scale Mass Spectrometry Data. *J. Proteome Res.* *20*, 2138-2144. 10.1021/acs.jproteome.0c00958.
20. Türei, D., Valdeolivas, A., Gul, L., Palacio-Escat, N., Klein, M., Ivanova, O., Ölbei, M., Gábor, A., Theis, F., Módos, D., et al. (2021). Integrated intra- and intercellular signaling knowledge for multicellular omics analysis. *Mol Syst Biol* *17*, e9923. 10.15252/msb.20209923.
21. Knight, J.D., Tian, R., Lee, R.E., Wang, F., Beauvais, A., Zou, H., Megeney, L.A., Gingras, A.C., Pawson, T., Figeys, D., and Kothary, R. (2012). A novel whole-cell lysate kinase assay identifies substrates of the p38 MAPK in differentiating myoblasts. *Skelet Muscle* *2*, 5. 10.1186/2044-5040-2-5.
22. Zoller, M.J., Nelson, N.C., and Taylor, S.S. (1981). Affinity labeling of cAMP-dependent protein kinase with p-fluorosulfonylbenzoyl adenosine. Covalent modification of lysine 71. *J Biol Chem* *256*, 10837-10842.
23. Derks, J., Leduc, A., Wallmann, G., Huffman, R.G., Willetts, M., Khan, S., Specht, H., Ralser, M., Demichev, V., and Slavov, N. (2022). Increasing the throughput of sensitive proteomics by plexDIA. *Nature biotechnology*. 10.1038/s41587-022-01389-w.
24. Hao, J., Cao, Y., Yu, H., Zong, L., An, R., and Xue, Y. (2021). Effect of MAP3K8 on Prognosis and Tumor-Related Inflammation in Renal Clear Cell Carcinoma. *Front Genet* *12*, 674613. 10.3389/fgene.2021.674613.
25. Suomalainen, M., Nakano, M.Y., Boucke, K., Keller, S., and Greber, U.F. (2001). Adenovirus-activated PKA and p38/MAPK pathways boost microtubule-mediated nuclear targeting of virus. *EMBO J* *20*, 1310-1319. 10.1093/emboj/20.6.1310.
26. Park, H.S., Kim, M.S., Huh, S.H., Park, J., Chung, J., Kang, S.S., and Choi, E.J. (2002). Akt (protein kinase B) negatively regulates SEK1 by means of protein phosphorylation. *J Biol Chem* *277*, 2573-2578. 10.1074/jbc.M110299200.
27. de Anda-Jauregui, G., Guo, K., McGregor, B.A., Feldman, E.L., and Hur, J. (2019). Pathway crosstalk perturbation network modeling for identification of connectivity changes induced by diabetic neuropathy and pioglitazone. *BMC Syst Biol* *13*, 1. 10.1186/s12918-018-0674-7.
28. Mendoza, M.C., Er, E.E., and Blenis, J. (2011). The Ras-ERK and PI3K-mTOR pathways: cross-talk and compensation. *Trends Biochem Sci* *36*, 320-328. 10.1016/j.tibs.2011.03.006.
29. Vert, G., and Chory, J. (2011). Crosstalk in cellular signaling: background noise or the real thing? *Developmental cell* *21*, 985-991. 10.1016/j.devcel.2011.11.006.
30. Frech, M., Andjelkovic, M., Ingley, E., Reddy, K.K., Falck, J.R., and Hemmings, B.A. (1997). High affinity binding of inositol phosphates and phosphoinositides to the pleckstrin homology domain of RAC/protein kinase B and their influence on kinase activity. *J Biol Chem* *272*, 8474-8481. 10.1074/jbc.272.13.8474.

31. Stokoe, D., Stephens, L.R., Copeland, T., Gaffney, P.R., Reese, C.B., Painter, G.F., Holmes, A.B., McCormick, F., and Hawkins, P.T. (1997). Dual role of phosphatidylinositol-3,4,5-trisphosphate in the activation of protein kinase B. *Science* *277*, 567-570. 10.1126/science.277.5325.567.
32. Truebestein, L., Hornegger, H., Anrather, D., Hartl, M., Fleming, K.D., Stariha, J.T.B., Pardon, E., Steyaert, J., Burke, J.E., and Leonard, T.A. (2021). Structure of autoinhibited Akt1 reveals mechanism of PIP(3)-mediated activation. *Proceedings of the National Academy of Sciences of the United States of America* *118*. 10.1073/pnas.2101496118.
33. Ebner, M., Lucic, I., Leonard, T.A., and Yudushkin, I. (2017). PI(3,4,5)P(3) Engagement Restricts Akt Activity to Cellular Membranes. *Molecular cell* *65*, 416-431 e416. 10.1016/j.molcel.2016.12.028.
34. Alessi, D.R., Deak, M., Casamayor, A., Caudwell, F.B., Morrice, N., Norman, D.G., Gaffney, P., Reese, C.B., MacDougall, C.N., Harbison, D., et al. (1997). 3-Phosphoinositide-dependent protein kinase-1 (PDK1): structural and functional homology with the Drosophila DSTPK61 kinase. *Curr Biol* *7*, 776-789. 10.1016/s0960-9822(06)00336-8.
35. Sarbassov, D.D., Guertin, D.A., Ali, S.M., and Sabatini, D.M. (2005). Phosphorylation and regulation of Akt/PKB by the rictor-mTOR complex. *Science* *307*, 1098-1101. 10.1126/science.1106148.
36. Lucic, I., Rathinaswamy, M.K., Truebestein, L., Hamelin, D.J., Burke, J.E., and Leonard, T.A. (2018). Conformational sampling of membranes by Akt controls its activation and inactivation. *Proceedings of the National Academy of Sciences of the United States of America* *115*, E3940-E3949. 10.1073/pnas.1716109115.
37. Chu, N., Viennet, T., Bae, H., Salguero, A., Boeszoermenyi, A., Arthanari, H., and Cole, P.A. (2020). The structural determinants of PH domain-mediated regulation of Akt revealed by segmental labeling. *eLife* *9*. 10.7554/eLife.59151.
38. Chu, N., Salguero, A.L., Liu, A.Z., Chen, Z., Dempsey, D.R., Ficarro, S.B., Alexander, W.M., Marto, J.A., Li, Y., Amzel, L.M., et al. (2018). Akt Kinase Activation Mechanisms Revealed Using Protein Semisynthesis. *Cell* *174*, 897-907 e814. 10.1016/j.cell.2018.07.003.
39. Kohn, A.D., Takeuchi, F., and Roth, R.A. (1996). Akt, a pleckstrin homology domain containing kinase, is activated primarily by phosphorylation. *J Biol Chem* *271*, 21920-21926. 10.1074/jbc.271.36.21920.
40. Johnson, J.L., Yaron, T.M., Huntsman, E.M., Kerelsky, A., Song, J., Regev, A., Lin, T.Y., Liberatore, K., Cizin, D.M., Cohen, B.M., et al. (2023). An atlas of substrate specificities for the human serine/threonine kinome. *Nature* *613*, 759-766. 10.1038/s41586-022-05575-3.
41. Gonzalez, E., and McGraw, T.E. (2009). Insulin-modulated Akt subcellular localization determines Akt isoform-specific signaling. *Proceedings of the National Academy of Sciences of the United States of America* *106*, 7004-7009. 10.1073/pnas.0901933106.
42. Sugiyama, M.G., Fairn, G.D., and Antonescu, C.N. (2019). Akt-ing Up Just About Everywhere: Compartment-Specific Akt Activation and Function in Receptor Tyrosine Kinase Signaling. *Front Cell Dev Biol* *7*, 70. 10.3389/fcell.2019.00070.
43. Jethwa, N., Chung, G.H., Lete, M.G., Alonso, A., Byrne, R.D., Calleja, V., and Larijani, B. (2015). Endomembrane PtdIns(3,4,5)P3 activates the PI3K-Akt pathway. *J Cell Sci* *128*, 3456-3465. 10.1242/jcs.172775.
44. Lien, E.C., Ghisolfi, L., Geck, R.C., Asara, J.M., and Toker, A. (2017). Oncogenic PI3K promotes methionine dependency in breast cancer cells through the cystine-glutamate antiporter xCT. *Sci Signal* *10*. 10.1126/scisignal.aao6604.
45. Siess, K.M., and Leonard, T.A. (2019). Lipid-dependent Akt-ivity: where, when, and how. *Biochem Soc Trans* *47*, 897-908. 10.1042/BST20190013.

REVIEWERS' COMMENTS

Reviewer #1 (Remarks to the Author):

Dear authors,

I appreciate that the authors comprehensively addressed by major and minor concerns in their rebuttal letter. First, the additional phospho-proteomics and cell viability assay ease out my concern about the specificity of optogenetic activation. Second, I believe the short-term perturbation may not impact protein expression is a valid point based on Fig S3. Last, the validated p-sites via western blots also strengthen the results by using other validation approach than MS-based proteomics, indicating the novel phosphoproteins are bona-fide Akt interaction partners. In summary, I think the revised version of this manuscript has been significantly improved the clarity of several key fundamental experiments, and the newer data provided in the rebuttal help to extend our understanding in Akt mediated signaling pathways. I would recommend accepting this manuscript with minor grammar edits.

Reviewer #2 (Remarks to the Author):

This reviewer thanks the authors for addressing adequately to the raised concerns, especially for the validation of their findings by the phosphoproteomics analysis. There were several typos in the revised figures and the text that should be checked by the authors.

1. The authors generally use the term "AKT", and use "Akt" when they refer to a specific isoform. However, sometimes it is not consistent. For example, on p.15 line 451, they state "... Akt1 downstream processes, in which AKT2 and AKT3 likely ...". Please recheck the usage of the term "AKT" throughout the text because inconsistent protein nomenclature might confuse the readers.

2. Several typos are found in the revised figures and text.

p.12, line 363: Should "342" (number of P-sites for PsPdb) also be italicized?

Figure 3F legend: “Aklt1” should be “Akt1”.

Fig. 2D: Should the enriched pathway name “E2F PATHWAY” for hC4 be changed to “E2F pathway”?

Fig. 3F: Although the revised figure seems appealing, the size of arrowheads and font sizes for proteins seems to be too small to read. In addition, can the authors make the consistent coloring of the cluster to Fig. 3E, if possible?

Fig. 6C: “5 inhibitors” should be “5 inhibitors”.

Figure S6B: y-axis label is missing.

Figure S10B, C: The axis label “Pearson_toAktT308” is not precise. They should be changed to something like, “Pearson’s R to AKT pT308”.

There is a spelling inconsistency in the term “PSPdb” and “PsPdb” throughout the text and figures.

Reviewer #3 (Remarks to the Author):

Summary (revision)

The authors have addressed the majority of my concerns in their revised manuscript. However, I feel that their discussion, particularly in which they compare their approach/dataset with other published datasets does not emphasize the technical advances made in this study. The authors are at pains to comment on the agreement between their dataset and others, while presenting clear evidence that the agreement is only partial at best (Supplementary Figure 10). In fact, their evidence is contrary to this notion – using more stringent criteria than simply motif enrichment, peptide array analysis, computational predictions, or pure observation, the authors present a highly curated list of experimentally validated substrates (direct or indirect) downstream of Akt. I think they could improve the discussion on pg 16 to describe precisely why their analysis will provide the field with a more stringent list of Akt substrate candidates. I appreciate that their cross-validation is important and that the overlap between datasets provides a greater degree of confidence in the designation of Akt substrates, but if all the authors presented was confirmation of what we already knew, it would not be of much utility to the field. This is a minor criticism, but I have the feeling that the work will gain more traction and be more impactful with a clear explanation of why the authors’ datasets should quite possibly be readers’ first port of call when making an initial assessment of whether their favorite protein may or may not be an Akt substrate.

Minor comments:

The authors have done a good job of addressing my concerns regarding their motif enrichment analysis. In particular, Figure S10E seems to me to be particularly important as it provides a framework for discriminating between substrates that bear a putative Akt consensus motif and those that also bear one, but have been additionally experimentally validated. The authors' comparison to the PhosphoSite Plus Database is informative, as it potentially identifies high-confidence versus erroneous sites. However, the figure itself could be improved with annotation of what is contained in the columns to the right of Ang-upregulated (presumably not upregulated by Ang) and those below Akt motif (presumably don't contain an Akt motif according to the definition in the figure legend).

The authors should clarify what is meant on pg 10 ln 364 by "(we identified 145 P-sites, hereafter PsPdb)." The PhosphoSitePlus database contains 342 reported Akt P-sites, but it seems that the authors only identified 145 of them in their analysis (though this is not 100% clear to me). This is an important clarification, as it implies that nearly 60% of the PSP P-sites are erroneous. The authors also provided a very useful motif analysis of their OptoCore list (Figure S10D) compared to the PSPdb list which reveals the enrichment of hydrophobic amino acids in the P+1 position. This both reinforces the greater stringency of their criteria for bona fide Akt substrates (112 OptoCore vs 342 PSPdb sites) and the original point made by this reviewer that Akt substrates have a preference for these amino acids at this position. It is, however, a shame that the authors do not comment on this observation in the main text of their manuscript, since this represents further refinement of what likely constitutes a bona fide Akt substrate motif.

The authors have confirmed the analysis of biological duplicates. Minor comment: pg 5 ln 160 seems to be missing the word 'and':

'Collectively, we used 27 conditions of AKT-activation (3 light intensities \times 3 patterns \times 3 time points), together with full darkness conditions as control, and (?) stimulated cells in biological duplicates for phosphoproteomic analysis.'

Supplementary Figure 1 legend. The legend to panel B is non-sensical and needs considerable revision:

Unfortunately, the authors' revision here is not an improvement. The figure legend is extremely poorly written. Furthermore, it conflates activity with Akt T308 phosphorylation, which is not correct. In fact, the data presented in S1B do not provide any measure of the activity of Akt, since no substrate is examined. T308 phosphorylation is not a proxy for Akt activity, especially in the context of a kinase-inactivating mutation. The figure legend needs revising and proper checking by a native English speaker before publication.

REVIEWERS' COMMENTS

Reviewer #1 (Remarks to the Author):

Dear authors,

I appreciate that the authors comprehensively addressed by major and minor concerns in their rebuttal letter. First, the additional phospho-proteomics and cell viability assay ease out my concern about the specificity of optogenetic activation. Second, I believe the short-term perturbation may not impact protein expression is a valid point based on Fig S3. Last, the validated p-sites via western blots also strengthen the results by using other validation approach than MS-based proteomics, indicating the novel phosphoproteins are bona-fide Akt interaction partners. In summary, I think the revised version of this manuscript has been significantly improved the clarity of several key fundamental experiments, and the newer data provided in the rebuttal help to extend our understanding in Akt mediated signaling pathways. I would recommend accepting this manuscript with minor grammar edits.

Author Reply: We thank the Reviewer for their recommendation. We have run the grammar check again and proofread the text carefully.

Reviewer #2 (Remarks to the Author):

This reviewer thanks the authors for addressing adequately to the raised concerns, especially for the validation of their findings by the phosphoproteomics analysis. There were several typos in the revised figures and the text that should be checked by the authors.

Author Reply: We highly appreciated this Reviewer for the careful read and for picking up the typos.

1. The authors generally use the term “AKT”, and use “Akt” when they refer to a specific isoform. However, sometimes it is not consistent. For example, on p.15 line 451, they state “... Akt1 downstream processes, in which AKT2 and AKT3 likely ...”. Please recheck the usage of the term “AKT” throughout the text because inconsistent protein nomenclature might confuse the readers.

Author Reply: Thanks for this comment. We are now advised by the journal to change “AKT” to “Akt1” wherever applicable. We agreed due to the particular optogenetic Akt1 system used. Therefore, “Akt1” is used throughout the manuscript. Of course, we used “AKT” for those places describing general biological pathways. We have now double-checked all the places and tried our best to stick to this principle.

2. Several typos are found in the revised figures and text.

p.12, line 363: Should “342” (number of P-sites for PsPdb) also be italicized?

Author Reply: Thanks. This is fixed.

Figure 3F legend: “Aklt1” should be “Akt1”.

Author Reply: Thanks for catching this! Corrected.

Fig. 2D: Should the enriched pathway name “E2F PATHWAY” for hC4 be changed to “E2F pathway”?

Author Reply: Yes. Changed accordingly.

Fig. 3F: Although the revised figure seems appealing, the size of arrowheads and font sizes for proteins seems to be too small to read. In addition, can the authors make the consistent coloring of the cluster to Fig. 3E, if possible?

Author Reply: Following this request, we have enlarged 200% the arrowheads and also enlarged the front sizes for all protein names so that the text labels are maximized in each node. The size of the text labels are not corresponding to the number of outgoing edges any more (whereas the node size was not changed and still corresponds to the number of outgoing edges). We hope this increases the readability of the network. Also, it is a great suggestion to keep the color consistent to the clusters in Figure 3E. We complied.

Fig. 6C: “5 inhibitors” should be “5 inhibitors”.

Author Reply: Thanks! Corrected.

Figure S6B: y-axis label is missing.

Author Reply: Thanks! Added.

Figure S10B, C: The axis label “Pearson_toAktT308” is not precise. They should be changed to something like, “Pearson’s R to AKT pT308”.

Author Reply: We complied.

There is a spelling inconsistency in the term “PSPdb” and “PsPdb” throughout the text and figures.

Author Reply: Thanks for pointing this out. We have now double checked and stick to “PSPdb”.

Reviewer #3 (Remarks to the Author):

Summary (revision)

The authors have addressed the majority of my concerns in their revised manuscript. However, I feel that their discussion, particularly in which they compare their approach/dataset with other published datasets does not emphasize the technical advances made in this study. The authors are at pains to comment on the agreement between their dataset and others, while presenting clear evidence that the agreement is only partial at best (Supplementary Figure 10). In fact, their evidence is contrary to this notion – using more stringent criteria than simply motif enrichment, peptide array analysis, computational predictions, or pure observation, the authors present a highly curated list of experimentally validated substrates (direct or indirect) downstream of Akt. I think they could improve the discussion on pg 16 to describe precisely why their analysis will provide the field with a more stringent list of Akt substrate candidates. I appreciate that their cross-validation is important and that the overlap between datasets provides a greater degree of confidence in the designation of Akt substrates, but if all the authors presented was confirmation of what we already knew, it would not be of much utility to the field. This is a minor criticism, but I have the feeling that the work will gain more traction and be more impactful with a clear explanation of why the authors’ datasets should quite possibly be readers’ first port of call when making an initial assessment of whether their favorite protein may or may not be an Akt substrate.

Author Reply: We thank this reviewer for appreciating the high quality of our dataset and their suggestions in improving the discussion to highlight the impact of the resource. We have strengthened the discussion accordingly. Please see **Line 494-498 Page 16**.

Minor comments:

The authors have done a good job of addressing my concerns regarding their motif enrichment analysis. In particular, Figure S10E seems to me to be particularly important as it provides a framework for discriminating between substrates that bear a putative Akt consensus motif and those that also bear one, but have been additionally experimentally validated. The authors' comparison to the PhosphoSite Plus Database is informative, as it potentially identifies high-confidence versus erroneous sites. However, the figure itself could be improved with annotation of what is contained in the columns to the right of Ang-upregulated (presumably not upregulated by Ang) and those below Akt motif (presumably don't contain an Akt motif according to the definition in the figure legend).

Author Reply: Thanks. The reviewer is correct in inferring the column/row heads. We have added "Not regulated by Ang" and "w/o AKT motif" in to the figure.

The authors should clarify what is meant on pg 10 ln 364 by "(we identified 145 P-sites, hereafter PsPdb)." The PhosphoSitePlus database contains 342 reported Akt P-sites, but it seems that the authors only identified 145 of them in their analysis (though this is not 100% clear to me). This is an important clarification, as it implies that nearly 60% of the PSP P-sites are erroneous. The authors also provided a very useful motif analysis of their OptoCore list (Figure S10D) compared to the PSPdb list which reveals the enrichment of hydrophobic amino acids in the P+1 position. This both reinforces the greater stringency of their criteria for bona fide Akt substrates (112 OptoCore vs 342 PSPdb sites) and the original point made by this reviewer that Akt substrates have a preference for these amino acids at this position. It is, however, a shame that the authors do not comment on this observation in the main text of their manuscript, since this represents further refinement of what likely constitutes a bona fide Akt substrate motif.

Author Reply: Sorry for not being clear about the numbers of P sites in PSPdb in our dataset. We are happy to make the clarification below.

Yes, the number of PSP P-sites is 342. However, the fact is that our phosphoproteomic measurement cannot detect and measure all of them: for example, some of them may not yield a tryptic peptide of a proper length to be detected by proteomics, or some of them might be too low in the abundance or stoichiometry to be detected by mass spectrometry, or some of them are not expressed in the cell type being analyzed. That is why we "detected" 145 P-sites in our present dataset. Therefore, our results cannot lead to the conclusion that 60% of the PSP P-sites are "erroneous". Our analysis simply do not cover that 60%. Needless to say, detecting 145 P sites listed in PSPdb (from the total of 34k phosphosites) is already very impressive. From Figure 6C, we can see that the current PSPdb P-sites are of high quality for sure.

We therefore changed text of "we identified 145 P-sites identified" to "145 of which were detected in our Phos-DIA experiment" (**Line 360 Page 12**). We also add a relevant note at **Line 491 Page 16**.

Regarding Figure S10D, following the reviewer's comments, we have mentioned the nice "P+1" result in the Result (main text). Please see **Line 377 Page 12**. "It is intriguing to note that majority of OptoCore P-sites tend to have hydrophobic amino acids at the P+1 position, in agreement with previous reports (**Supplementary Fig. 10d**)".

The authors have confirmed the analysis of biological duplicates. Minor comment: pg 5 ln 160 seems to be missing the word 'and':

'Collectively, we used 27 conditions of AKT-activation (3 light intensities $\text{\AA} \sim 3$ patterns $\text{\AA} \sim 3$ time points), together with full darkness conditions as control, and (?) stimulated cells in biological duplicates for phosphoproteomic analysis.'

Author Reply: Thanks. This sentence is now rewritten as "...together with full darkness conditions as a control, for phosphoproteomic analysis, with all conditions examined in biological duplicates."

Supplementary Figure 1 legend. The legend to panel B is non-sensical and needs considerable revision:

Unfortunately, the authors' revision here is not an improvement. The figure legend is extremely poorly written. Furthermore, it conflates activity with Akt T308 phosphorylation, which is not correct. In fact, the data presented in S1B do not provide any measure of the activity of Akt, since no substrate is examined. T308 phosphorylation is not a proxy for Akt activity, especially in the context of a kinase-inactivating mutation. The figure legend needs revising and proper checking by a native English speaker before publication.

Author Reply: This is a fair advice. Following this comment, we have now tried our best to re-write the Supplementary Figure 1 legend, especially the one for Figure S1B. In particular, we removed the term of “the activity of Akt” but just use “AKT Thr308 phosphorylation” in the legend. Please see the rewritten legend below:

Figure S1. Establishment of the optogenetic cell culture system for inducing and controlling Akt phosphorylation.

(a) A manually engineered light system in cell culture incubator, which has a blue LED light (460-470 nm) plugged into a timer, a lifting table, a filter and a diffuser between the light and the sample to achieve proper and uniform light exposure. The timer can control the light exposure time from 1 minute to 24 hours and set programs to achieve periodic light illumination. The lifting table can change the distance between the light and the sample to adjust the light magnitude reached the sample. (b) Immunoblot comparing phosphorylation of exogenous optogenetic wild-type Akt1 and Akt1 R25C and Akt1 K179M mutants in response to 20% FBS or blue light illumination. Akt1 R25C carried a mutation in the PH domain that blocks recruitment to the plasma membrane, preventing phosphorylation and or further activation. Akt1 K179M has a kinase inactivating mutation in the catalytic domain. The immunoblot shows that the optogenetic wild-type Akt1 could be phosphorylated by serum or blue light stimulation, while Akt1 R25C was weakly phosphorylated. Kinase inactive Akt1 K179M was phosphorylated to a lower extent than wild-type Akt1. Endogenous Akt1 was phosphorylated following serum stimulation regardless of the light condition.